# Convective Heat Transfer of Spring Meltwater Accelerates Active Layer Phase Change in Tibet Permafrost Areas

Yi Zhao[1], Zhuotong Nan[1,2*], Hailong Ji[1], Lin Zhao[3]

5    [1]Key Laboratory of Ministry of Education on Virtual Geographic Environment, Nanjing Normal University, Nanjing, 210023, China

[2]Jiangsu Center for Collaborative Innovation in Geographical Information Resource Development and Application, Nanjing, 210023, China

[3]School of Geographical Sciences, Nanjing University of Information Science & Technology, Nanjing

10   210044, China

*Corresponding author*: Zhuotong Nan (nanzt@njnu.edu.cn)

**Abstract**

Convective heat transfer (CHT) is one of the important processes that control the near ground surface heat transfer in permafrost areas. However, this process has often not been considered in most permafrost studies, and its influence on freezing-thawing processes in the active layer lacks quantitative investigation. The Simultaneous Heat and Water (SHAW) model, one of the few land surface models in which the CHT process is well incorporated into the soil heat-mass transport processes, was applied in this study to investigate the impacts of CHT on the thermal dynamics of the active layer at the Tanggula station, a typical permafrost site at the eastern Qinghai-Tibet Plateau with abundant meteorological and soil temperature/soil moisture observation data. A control experiment was carried out to quantify the changes in active layer temperature affected by vertical advection of liquid water. Three experimental setups were used: (1) the original SHAW model with full consideration of CHT; (2) a modified SHAW model that ignores CHT due to infiltration from the surface, and (3) a modified SHAW model that completely ignores CHT processes in the system. The results show that the CHT events occurred mainly during thaw periods in melted shallow (0-0.2 m) and intermediate (0.4-1.3 m) soil depths, and their impacts on soil temperature at shallow depths were significantly greater during spring melting periods than summer. The impact was minimal during freeze periods and in deep soil layers. During thaw periods, temperatures at the shallow and intermediate soil depths simulated under the scenario considering CHT were on average about 0.9 and 0.4 °C higher, respectively, than under the scenarios ignoring CHT. The ending dates of the zero-curtain effect were substantially advanced when CHT was considered due to its heating effect. However, the opposite cooling effect was also present but not as frequently as heating, due to upward liquid fluxes and thermal differences between soil layers. In some periods, the advection flow from the cold layer reduced the shallow and intermediate depth temperatures by an average of about -1.0 and -0.4 °C, respectively. The overall annual effect of CHT due to liquid flux is to increase soil temperature in the active layer and favor thawing of frozen ground at the study site.

**Keywords:** convective heat transfer, active layer, permafrost, hydrological and thermal processes, Simultaneous Heat and Water (SHAW) Model

## 1. Introduction

Permafrost is defined as the ground that remains frozen consecutively for more than two years (Zhao et al., 2010), and is mainly distributed at high latitudes and cold alpine areas, such as the Antarctic, Arctic and Qinghai-Tibet Plateau (QTP) (Zhang et al., 1999). Given the current warming trends in most of the Earth's permafrost areas (Biskaborn et al., 2019), significant changes in permafrost dynamics are likely to occur, and the local ecosystem and environment have already been seriously influenced by regional hydrological and thermal changes caused by permafrost degradation (Cheng and Wu, 2007; Jin et al., 2009; Jorgenson et al., 2001; Tesi et al., 2016). It is thus very essential to understand thoroughly the thermal and hydrological processes in the soil in frozen ground regions.

It is generally recognized that ground heat transfer is more than a single heat conduction process controlled by upper and lower boundary conditions, but a complex system that accounts for both conductive and non-conductive heat transfer (Kane et al., 2001; Putkonen, 1998). Non-conductive heat processes refer to all those heat transfer processes that can significantly impact the thermal regime but not explicitly described by heat conduction theory, including: (1) latent heat exchange; (2) vapor convective heat transfer (CHT) caused by vapor pressure gradients or thermal gradients (Cahill and Parlange, 1998); (3) CHT due to infiltration of snowmelt water and rainwater from surface and due to advection within soils (Scherler et al., 2011; Woo et al., 2000). Despite the predominance of thermal conduction in permafrost regions, the role of non-conductive processes on the freeze-thaw cycles of the active layer cannot be ignored (Kane et al., 2001). Vapor fluxes between soils due to temperature and pressure gradients (Cahill and Parlange, 1998; Halliwell and Rouse, 1987) and soil moisture evaporation (Roth and Boike, 2001; Shen et al., 2015) usually exert a cooling effect on soil thermal regimes, which is used to protect engineering infrastructure from frost heave damage in permafrost regions (Cheng, 2004; Cheng et al., 2008). Migration of liquid water can usually be forced by gravitational, pressure or osmotic pressure gradients in soils during thaw periods. During spring snowmelt and summer rainfall, a rapid temperature increase of about 2°C to 4°C is observed in the uppermost soil layer, indicating a heating effect of liquid CHT (Hinkel et al., 1996; Hinkel et al., 1997; Kane et al., 1991). As a result, warming of soil temperature by liquid CHT due to summertime rainfall increases the thaw depth of frozen ground (Douglas et al., 2020; Guan et al., 2010; Karjalainen et al., 2019) and promotes the greenhouse gas emissions (Neumann et al., 2019). However, an opposite view also exists that infiltration of precipitation

has a cooling effect on the temperature of the active layer (Wen et al., 2014; Yang et al., 2018), indicating
a complex mechanism of the CHT impacts on the soil thermal regime. In freeze periods, residual water convection could ensue and the CHT of liquid water is relatively modest but still works, because the freezing process occurring in the active layer increases the pore fluid density and van der Waals forces on the ice particles surface (Fisher et al., 2020; Kane and Stein, 1983).

Understanding the impact of CHT on frozen ground is important for accurate simulation of ground temperature and associated hydrology in permafrost regions in the context of global climate warming. Although some convective heat effects have been observed, they are often produced by simultaneous processes such as heat conduction, advection and convection, and phase changes. In-situ instrumentation is still limited to accurately measure key thermal and hydrological soil variables. It is challenging to isolate the sole impacts of CHT from the totality of heat transfer processes in the soil (Hasler et al., 2008; Pogliotti et al., 2008).

Physically explicit numerical models are effective tools for isolating the impact of a single process from the overall system. These models could provide information that is otherwise impossible through observation techniques. Most existing land surface models such as Noah land surface model (LSM) and Community Land Model (CLM) only account for heat conduction and phase change in the energy budget, despite their extensive use in modelling processes in cold regions (Gao et al., 2019; Guo and Wang, 2013; van der Velde et al., 2009; Wu et al., 2018). Neglecting the CHT mechanism in these models generally leads to increased uncertainty due to physical inadequacies. The demand for complete modelling of permafrost changes has therefore recently prompted interest in the development of simulation tools for coupled heat transport and variable hydrological processes to account specifically for CHT. A number of traditional schemes for soil heat transport have been further developed with enhanced vapor/liquid CHT processes and have been shown to be effective in cold regions (He et al., 2018; Kurylyk et al., 2014; Wang and Yang, 2018). Furthermore, researchers have recently begun to formulate soil heat and water transport processes within a three-dimensional framework to provide a more reasonable physical expression for vertical and horizontal heat and mass transport (Orgogozo et al., 2019; Painter et al., 2016). By using these advanced models, the role of CHT on the permafrost thermal regime, especially the vapor CHT, was provisionally explained. Wicky et al. (2017) developed a numerical model considering air flow in permafrost talus slopes and revealed pronounced seasonality of the air flow cycle on talus slopes

and considerable seasonal differences in the effects on soil temperature. Yu et al. (2018) and Yu et al. (2020) quantified the thermal responses to different types of vapor migration associated with evaporation and air flow, respectively. Luethi et al. (2017) estimated the heat transfer efficiency of vapor and liquid convection. However, relatively few studies have examined liquid CHT processes in permafrost context. Kurylyk et al. (2016) developed a three-dimensional coupled soil heat and water model to investigate the effects of runoff on soil temperature. Recently, Zhang, M. et al. (2021) quantified the energy flux of infiltrative CHT during a summertime rainfall event and reported that the thermal impacts of CHT were not pronounced compared to other energy transfer pathways. While their studies improve our understanding on the role of CHT in altering permafrost thermal dynamics, they focused on specific permafrost conditions or single events, and the established methods were difficult to transfer to other regions with conditions dissimilar to those in these study regions.

The Simultaneous Heat and Water (SHAW) model is one of the well-known one-dimensional coupled hydraulic-thermal models that integrates mass and energy transfer processes of the atmosphere-vegetation-soil continuum into a simultaneous solution (Flerchinger and Saxton, 1989a). The SHAW model is one of the few land surface models (LSMs) that considers the detailed physics of the interrelated mass and energy transfer mechanisms, including precise convective heat transport processes of liquid water and vapor (Kurylyk and Watanabe, 2013), making it advantageous for demonstrating the important interactions between soil water dynamics and frozen soil thermal regimes in permafrost regions (Flerchinger et al., 2012). In addition, SHAW applies a special iteration scheme in which a time step is subdivided into multiple sub-time steps to control the error from the previous step in solving the mass and energy balance and to strictly enforce the mutual coupling of the hydrological and thermal processes (Flerchinger, 2000). The SHAW model has many applications in permafrost regions, including the studies of permafrost hydrological and thermal processes (Chen et al., 2019; Cui et al., 2020), permafrost evolution (Wei et al., 2011) and frozen ground responses to climate and land ecosystem changes (Huang and Gallichand, 2006; Kahimba et al., 2009; Link et al., 2004; Zuo et al., 2019). Previous studies has indicated that SHAW can simulate the dynamics of soil temperature, soil moisture (Flerchinger and Pierson, 1997) and the freeze-thaw cycles (Flerchinger and Saxton, 1989b) that occur repeatedly in the active layer with good accuracy. The fine consideration of CHT processes, the mutual coupling of

hydrothermal processes, and the broad applicability render the SHAW model capable of investigating the impacts of CHT on permafrost thermal regimes.

Therefore, this study uses the SHAW model to quantify the impacts of liquid CHT on the soil temperature and moisture in the active layer through numerical modelling at a typical permafrost site, i.e., the Tanggula (TGL) site on the QTP, China. The SHAW model was modified to exclude the CHT processes, and then control experiments were implemented to simulate comparative scenarios with or without CHT included in the model. This enables precise and separate quantification of the thermal impacts of liquid CHT from different sources such as precipitation infiltration, snow melt and ground ice melt, which has never been accomplished in previous studies. The specific objectives are: (1) to illustrate the characteristics of CHT events in time and depth; (2) to quantify the sole impacts of liquid CHT on the thermal regime of the active layer; (3) to elucidate the interplay of heat and soil moisture during the freezing-thawing process in the active layer.

## 2.    Methods and Data

### 2.1 Mathematical representation in the Simultaneous Heat and Water model

The SHAW model was developed to simulate heat, water and solute transfer within a one-dimensional profile which includes the effects of plant cover, snow, dead plant residue and soil. The model stratifies the soil column into soil layers. For each soil layer, the net energy budget is equal to the sum of conductive heat flux, CHT from liquid and vapor migration, latent heat from water phase change, and temperature change. The one-dimensional energy balance equation for each layer is:

$$\frac{\partial}{\partial z}\left(k_s \frac{\partial T}{\partial z}\right) - \rho_l c_l \frac{\partial(q_l T)}{\partial z} - L_v \left(\frac{\partial \rho_v}{\partial t} + \frac{\partial q_v}{\partial z}\right) = C_s \frac{\partial T}{\partial t} - \rho_i L_f \frac{\partial \theta_i}{\partial t} \qquad (1)$$

where $C_s$ is the effective volumetric heat capacity of the soil layer (J·m$^{-3}$·°C$^{-1}$), which is a lumped influence of minerals, liquid, ice, and vapor in the soil layer; $T$ is the soil temperature in that layer (°C); $\rho_i$, $\rho_l$, $\rho_v$ are the densities of ice, liquid water, and vapor (kg·m$^{-3}$), respectively; $L_f$ and $L_v$ are the latent heats of fusion and vapor (kJ·kg$^{-1}$), respectively; $\theta_i$ is the volumetric ice content of the soil layer (m$^3$·m$^{-3}$); $q_l$ and $q_v$ are the liquid water flux (m·s$^{-1}$) and vapor flux (kg·m$^{-2}$ s$^{-1}$), respectively; $k_s$ is the soil thermal conductivity (W·m$^{-1}$·°C$^{-1}$); $c_l$ is the specific heat capacity of water (J·kg$^{-1}$·°C$^{-1}$); $t$ and $z$ are the time step and the depth at the midpoint of the current layer, respectively. The second term on the left-

hand side of Eq. 1 represents the heat flux caused by the migration of liquid pore water. The SHAW

model assumes that the migrating liquid and vapor water fluxes have the same temperature as the layers

in which they are generated. Since the model does not provide a specific estimate of rain temperature

and ignores the CHT processes within the canopy layer, rainwater entering the residue layer through the

canopy is simply assumed to be at the same temperature as the residue layer when no snow cover is

present on the surface. When snow is present, rainwater flowing through the canopy will participate in

snow processes before it reaches the residue layer, and the temperature of snowmelt is assumed to be the

same as the temperature of the snow layer at the time of melting.

The diffusion equation for the soil moisture of each layer is:

$$\frac{\partial}{\partial z}\left[K\left(\frac{\partial \psi}{\partial z}+1\right)\right]+\frac{1}{\rho_l}\frac{\partial q_v}{\partial z}+U=\frac{\partial \theta_l}{\partial t}+\frac{\rho_i}{\rho_l}\frac{\partial \theta_i}{\partial t} \tag{2}$$

where $K$ is the unsaturated hydraulic conductivity (cm·h$^{-1}$); $\psi$ is the soil matric potential (m), and $U$ is a

source/sink term for water uptaken by roots (m$^3$·m$^{-3}$·s$^{-1}$). $K$ is determined by:

$$K=K_s\left(\frac{\Psi_e}{\Psi}\right)^{\left(2+\frac{3}{b}\right)} \tag{3}$$

where $K_s$ is the saturated hydraulic conductivity (cm·h$^{-1}$), $b$ is an empirical parameter representing pore

size distribution, and $\Psi_e$ is the air entry potential (m) for the saturated soil layer. $\psi$ is computed as a

function of soil moisture:

$$\Psi=\Psi_e\left(\frac{\theta_l}{\theta_s}\right)^{-b} \tag{4}$$

where $\theta_l$ and $\theta_s$ are the liquid water content (m$^3$·m$^{-3}$) and soil porosity (m$^3$·m$^{-3}$), respectively. Then, the

vertical water flux $q_l$ could be calculated by the water potential difference and the relative conductivity

$K_{n,n+1}$ between layer $n$ and $n+1$:

$$K_{n,n+1}=(K_n \cdot K_{n+1})^{\frac{1}{2}} \tag{5}$$

$$q_l=\frac{K_{n,n+1}}{z_{n+1}-z_n}(\Psi_n-\Psi_{n+1}+z_{n+1}-z_n) \tag{6}$$

If the soil temperature is below 0 ° C and ice is present, the total soil water potential is estimated using

a modified Clausius-Clapeyron equation:

$$\emptyset=\Psi+\pi=\frac{L_f}{g}\left(\frac{T}{T_K}\right) \tag{7}$$

where $\emptyset$ is the total soil water potential, $\Psi$ is the matric potential from Eq. 4, $\pi$ is the osmotic potential

(m) with respect to solutes in the soil, $g$ is the acceleration due to gravity (m·s$^{-2}$), $L_f$ is the latent heat

consumption during phase change (kJ·kg$^{-1}$), and $T_K$ is the freezing point (°C). Thus, the unfrozen water content and ice content can be solved by combining Eq. 4 and Eq. 7:

$$\theta_l = \theta_s \left( \left( \frac{L_f T}{g T_k} - \pi \right) / \Psi_e \right)^{-\frac{1}{b}} \tag{8}$$

$$\theta_i = (\theta_w - \theta_l) \frac{\rho_i}{\rho_l} \tag{9}$$

where $\theta_w$ is the total water equivalent in the soil layer. When ice content is present, hydraulic conductivity is inhibited:

$$K_i = \begin{cases} 0, & p - \theta_i < 0.13 \\ f \cdot K, & p - \theta_i \geq 0.13 \end{cases} \tag{10}$$

where $p$ is the available porosity, and $f$ is a fraction for linearly reducing the soil hydraulic conductivity:

$$f = \frac{p - \theta_i - 0.13}{p - 0.13} \tag{11}$$

Inputs of the SHAW model consist of three types of data: (1) meteorological forcing data, including air temperature, relative humidity, wind speed, precipitation, new snow density, and shortwave radiation; (2) soil moisture content and soil temperature data as initial conditions and lower boundary conditions; (3) characteristic parameters of vegetation canopy, snow, dead plant residue, and soil column at the study site.

**2.2 Design of the control experiment**

The SHAW model incorporates the CHT processes of the liquid and vapor flux into the energy conservation equation, making it possible to portray complete water-heat interactions that frequently occur in the freezing-thawing processes in permafrost regions. In this study, we designed a control experiment with three scenarios to represent the full presence, partial presence and full absence of liquid CHT in the model by modifying the model codes. The same forcing data at a typical permafrost site, the TGL, and the same parameter values were used for the three modified models. The impacts of liquid CHT on the active layer dynamics are quantified as differences in soil temperature and moisture content by contrasting the results of the three scenarios.

The control experiment consists of three scenarios:

1. Control: In this setup, the original SHAW model is applied to the TGL site and the simulated results serve as a baseline to contrast with the results of the other scenarios. Confined within the

physical limits, the soil parameters for each layer were calibrated to best match the simulated soil temperatures and moisture contents with the observations at different depth, i.e., 0.05, 0.1, 0.4, 1.05 and 2.45 m. The same calibrated soil parameter values are used in the other two scenarios to ensure consistency throughout the experiment.

2.  No surface CHT (NoSurf): The CHT between the ground surface and the soil is not considered in this setup. For this purpose, the codes related to liquid water CHT from the ground surface to the top soil layer (0.00 m), as described in the second term on the left-hand side of Eq. 1 were disabled in the modified SHAW model. By contrasting the model outputs of this setup with those of Control, the effects of the infiltrative convective heat could be quantified.

3.  No CHT (NoConv): In this scenario, the liquid water CHT is completely eliminated, i.e., both the infiltrative convection from the surface to the top soil and the heat transfer associated with the liquid water migration within the soil layers are not considered. All codes related to the second term on the left-hand side of Eq. 1 were disabled for the whole soil column. By contrasting Control−NoConv with Control−NoSurf, the impacts of CHT relating to vertical

advection within soil layers are demonstrated.

Note that in the NoSurf and NoConv setups, we removed only the heat fluxes and exchanges associated with water movement and retained water movement itself, which is necessary to maintain the water balance in each soil layer. In the SHAW model, we found that the simulated direction of vapor flux did not match well the real vapor cycle, so the convection associated with the vapor remained intact in the

three setups to exclude the impacts of vapor CHT in this analysis. The resulting differences between the NoSurf/NoConv and Control simulations, each configured by the same model settings including the same meteorological forcing data, lower boundary conditions, and calibrated parametric values, thus reflect the effects of liquid CHT on the active layer dynamics. Higher simulated soil temperatures in the Control than in the other two scenarios imply a positive thermal impact of CHT on the active layer, and if lower,

a negative one. We defined a CHT event as a ground temperature deviation of more than 0.1 °C between NoSurf/NoConv and Control at one model time step. A deviation of 0.1 °C or less is trivial and could be due to model iteration bias rather than CHT. According to their effect, there are cooling CHT events and heating CHT events. The total numbers of cooling and heating CHT events and mean temperature deviations were analyzed to examine the frequency and magnitude of CHT effects on ground temperature:

$$\overline{\Delta T} = \frac{\sum_{i=1}^{m} \Delta T}{m} \qquad (12)$$

where, $\overline{\Delta T}$ is the mean temperature deviation of all heating or cooling CHT events, $\Delta T$ is the temperature deviation caused by an CHT event, and $m$ indicates the count of heating or cooling CHT events.

## 2.3 Experimental area and data

A typical permafrost site, the TGL site on the QTP, was selected for this study because of long-term,
quality-assured observations of the active layer and deep permafrost in parallel with meteorological observations at high temporal resolution. Due to the ideal representativeness to elevation-controlled permafrost on the QTP, this site has been widely used for alpine permafrost research such as permafrost hydrothermal characteristics (Li et al., 2019), permafrost response to climate change (Zhu et al., 2017; Zhu et al., 2021), and permafrost process modelling (Hu et al., 2015; Li et al., 2020). The TGL site (33°04′
N, 91°56′ E) is situated on a southwest-facing slope elevated at 5100 m above sea level (a.s.l.) in the Tanggula mountains on the eastern QTP (Figure 1b). The local vegetation is sparse alpine meadow with a coverage fraction of about 30~40%. Soils are mainly loamy sand (sand content >70%). The annual mean air temperature is about -4.9 °C. The active-layer thickness (ALT) is measured to be about 3 m (Xiao, Zhao, and Dai et al., 2013). About 400 mm of precipitation falls per year, mainly concentrated in
the months of May to September, accounting for 92% of the total year. According to continuous snow depth monitoring by an SR-50 ultrasonic snow depth sensor, the instantaneous maximum snow depth in the vicinity of the TGL site is about 22 cm, and the days with snow depth below 5 cm account for 72% of all snow days (Xiao, Zhao, and Li et al., 2013).

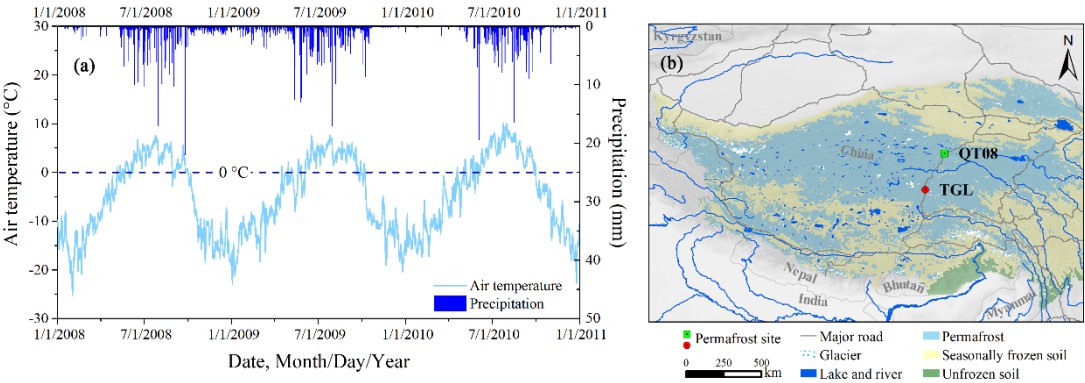

**Figure 1  Air temperature and precipitation (a) at the Tanggula (TGL) site during 2008-2010 and the map (b) showing the locations of permafrost sites considered in this study. Times series of daily air temperature at 2**

**m height and precipitation at TGL aggregated from the hourly measurements. The base map of permafrost distribution on the Qinghai-Tibet plateau (QTP) is from Zou et al. (2017) .**

Installed instruments include an automatic weather station that measures air temperature, wind speed and direction, humidity, shortwave/longwave radiation (upward and downward), air pressure, snow depth, and precipitation, and an active-layer monitoring system that measures soil temperatures and moisture contents at the depths of 0.05, 0.1, 0.2, 0.4, 0.7, 1.05, 1.3, 1.75, 2.1, 2.45 and 2.8 m below the surface. The time series of half-hourly air temperature, relative humidity, wind speed, precipitation and shortwave radiation at 2 m collected from the automatic weather station at the TGL site from 2008 to 2010 were used to run the SHAW model with a time step of one hour (Figure 1a). In the SHAW model, precipitation is assumed to be snowfall when air temperature is below 1 °C. The observed daily soil temperature and unfrozen water content (UWC) at 0.05, 0.1, 0.4, 1.05 and 2.45 m depth during the same period collected from the active layer monitoring system were used to calibrate and validate the SHAW model.

**2.4 Model settings**

In addition to the hourly meteorological data from the TGL site as driving data, the inputs to the SHAW model also include the snow density of each new snowfall event. We set them as zeros and let the model estimate them based on the air temperature at the time.

The soil column was stratified into 13 layers corresponding to the observation depths at the TGL site, including five layers (centered at 0.00, 0.02, 0.05, 0.10 and 0.20 m) as shallow depths, five layers (centered at 0.40, 0.50, 0.70, 1.05 and 1.30 m) as intermediate depths, and three layers (centered at 1.75, 2.10 and 2.45 m) as deep depths. The SHAW model defines a special soil layer at the depth of zero that accounts for heat and water exchange at the interface between the plant residue layer and the soil. The shallow soil depths were densely discretized to accommodate rapid hourly variations in soil temperature and moisture near the ground surface. Table 1 shows the vertical discretization of the TGL soil profile and the measured volumetric fractions of sand, silt and clay and the bulk density for each layer.

The SHAW model depends on accurate lower boundaries, which are usually specified at a shallow depth to allow accurate simulation of coupled water-heat exchange processes (Chen et al., 2019). In this study, observations at a depth of 2.8 m near the bottom of the active layer were set as lower boundaries. The observed daily soil temperatures at this depth constrain the heat fluxes through the lower boundary. The

lower boundary of soil moisture content (both ice and UWC) was determined by the model following an

empirical equation as a function of soil temperature by confining the liquid water equivalent maxima of

$0.25 \text{ m}^3 \cdot \text{ m}^{-3}$ that occur in summer.

For all scenarios, initial soil temperature and soil moisture profiles were generated with a three-decade

spin-up using forcing data cycling from 2008-2010 until differences in soil temperature and moisture

content are reduced to less than $0.1 \text{ °C}$ and $0.01 \text{ m}^3 \cdot \text{ m}^{-3}$, respectively, between the last and penultimate

cycles at the same time for all soil layers. The final soil temperature and soil moisture profiles were

provided as initial conditions to each scenario simulation.

According to a previous study at the same TGL site (Liu et al., 2013), the SHAW model with the default

parameter values simulated surface energy fluxes and soil temperatures well, except for soil moisture,

which was seriously underestimated. We calibrated the four main hydraulic parameters (Table 1), i.e.,

saturated hydraulic conductivity, air-entry potential, saturated volumetric moisture content, and pore-size

distribution index, relating to soil moisture in the model, while keeping the other soil parameters as

default values. Data from 2008-2009 were used for calibration and 2010 for validation. The model was

run with an hourly time step and the results were then aggregated to a daily scale to facilitate comparisons

and analyses. The ranges of hydraulic parameter values were roughly determined with reference to

previously studies (Chen et al., 2019; Liu et al., 2013; Wu et al., 2018). To find the best parameter

combination and measure model uncertainty, 1 000 independent parameter combinations randomly

generated by the Latin hypercube sampling method in conjunction with the priori ranges. We restricted

the values of sampling parameters in adjacent layers to assume that adjacent soil layers have similar

textures. Then the 1000 combinations were used to drive the model one by one, and their outputs were

compared and evaluated to determine the optimal parameter values for each soil layer. Two metrics,

including the Nash-Sutcliffe efficiency coefficient (NSE) and root mean square error (RMSE), were used

to quantify the performance of the parameter combinations:

$$\text{NSE} = 1 - \frac{\sum_{t=1}^{N}(O^t - M^t)^2}{\sum_{t=1}^{N}(O^t - \bar{O})^2} \tag{13}$$

$$\text{RMSE} = \sqrt{\frac{1}{N}\sum_{t=1}^{N}(O^t - M^t)^2} \tag{14}$$

where $O^t$ and $M^t$ are the observed value and simulated value at time step $t$; $\bar{O}$ is the mean of the

observations over the entire period; and $N$ is the total number of time steps. Considering the interaction

between soil temperature and soil moisture in a coupled system, the simulation accuracy of both variables

is mutually suppressed, i.e., while the accuracy of one variable continues to improve by continuously

optimizing its parameter value, the accuracy of the other decreases. Thus, we determined the optimal

parameter combinations by balancing the performances for both soil temperature and moisture. In

addition, the 95% probability bands (95PPU) of simulated soil temperature and moisture of all 1000

random parameter combinations were also counted, showing the range of distribution of results due to

parameter degrees of freedom, to measure model uncertainty introduced by parameter selection at the

TGL site.

The most optimal parameter values from the 1000 combinations, as presented in Table 1, were

consistently applied to all three scenarios designed in Section 2.2 to eliminate the influence of parameter

values on the inter-scenario comparison.

**Table 1  Key soil parameter values for the TGL soil profile; $\rho_b$, bulk density; $K_s$, saturated hydraulic conductivity; $\psi_e$, air-entry potential, $\theta_s$, saturated volumetric moisture content; $b$, pore-size distribution index.**

| Depth (m) | Sand (%) | Silt (%) | Clay (%) | $\rho_b$ (kg·m⁻³) | $K_s$ (cm·h⁻¹) | $\psi_e$ (m) | $\theta_s$ (m³·m⁻³) | $b$ |
|---|---|---|---|---|---|---|---|---|
| 0.00 | 93 | 1 | 6 | 1176 | 25.5 | -0.5 | 0.35 | 4.74 |
| 0.02 | 93 | 1 | 6 | 1176 | 25.5 | -0.5 | 0.35 | 4.74 |
| 0.05 | 93 | 1 | 6 | 1176 | 25.5 | -0.5 | 0.35 | 4.74 |
| 0.10 | 93 | 1 | 6 | 1176 | 25.5 | -0.5 | 0.35 | 4.74 |
| 0.20 | 87 | 3 | 10 | 1331 | 25.5 | -0.5 | 0.35 | 4.26 |
| 0.40 | 89 | 2 | 9 | 1103 | 25.5 | -0.3 | 0.3 | 4.26 |
| 0.50 | 87 | 3 | 10 | 1405 | 25.5 | -0.3 | 0.3 | 4.26 |
| 0.70 | 84 | 3 | 13 | 1405 | 20.05 | -0.3 | 0.3 | 4.26 |
| 1.05 | 75 | 7 | 18 | 1235 | 20.05 | -0.3 | 0.3 | 3.88 |
| 1.30 | 75 | 7 | 18 | 1281 | 20.05 | -0.2 | 0.3 | 3.88 |
| 1.75 | 71 | 8 | 21 | 1253 | 20.05 | -0.2 | 0.3 | 3.88 |
| 2.10 | 71 | 8 | 21 | 1460 | 20.05 | -0.2 | 0.3 | 3.88 |
| 2.45 | 71 | 8 | 21 | 1332 | 20.05 | -0.2 | 0.3 | 3.88 |

## 3. Results

### 3.1 Model evaluation

Figure 2 shows the SHAW simulations of soil temperature (left panels) and UWC (right panels) with the most optimal parameters and the observations at depths of 0.05, 0.1, 0.4, 1.05 and 2.45 m at the TGL site, as well as the 95PPU of model outputs as determined by all 1000 random parameter combinations. Overall, both the 95PPUs and the optimal outputs confirm a good capability of the SHAW model to simulate the complex freezing and thawing processes in the active layer given reliable lower boundaries.

Seasonal variations of both soil temperature and soil moisture in the active layer of the TGL were successfully captured. The 95PPUs of soil temperature associated with the 1000 parameter combinations are narrow in band and cover the observations well at each depth, indicating the good performance and low uncertainty of the SHAW model in modelling soil temperature at the TGL site. According to our experiments, saturated hydraulic conductivity is the most important parameter that effects the simulated

soil temperature. Although the 95PPUs of the simulated UWC also roughly cover the observations, a wide band and overestimation at 0.4 and 1.05 m depths relative to the observations indicate a large uncertainty in simulating UWC and call for a necessary parameter calibration. Saturated hydraulic conductivity and saturated volumetric moisture content were identified as the most important parameters controlling simulated UWC and were treated carefully. At the intermediate depths where low liquid

contents were observed, optimal parameter values are picked from the random parameter combinations for these layers that both simulate lower UWC and ensure good accuracy of the simulated soil temperature. The simulated soil temperatures with the optimal parameter combination were in particularly good agreement with the observed temperatures at the TGL site during both the calibration and validation periods (Figure 2). Specifically, the NSE values between the simulated and observed soil

temperatures are above 0.70 in most soil layers in both periods, except for the value at 1.05 m depth of in the validation period, and are highest (up to 0.90) in the shallow layers. The RMSE values for soil temperature decrease with soil depth because there is less interannual variation in the deep layers than in the shallow layers. Despite the relatively lower performance in UWC, simulation of UWC with optimal parameters still produced NSE values greater than 0.42 in all soil layers, and RMSE values of about 0.05

$m^3 \cdot m^{-3}$. During the summer of 2009, we noted an abrupt decline in observed UWC at 0.05 and 0.1 m

depths (Figure 2f, g), which was due to equipment malfunction. At depths of 0.4 and 1.05 m, some unrealistic zero UWC values were also observed during the winter months (Figure 2h, i). Many studies have already affirmed that a small amount of liquid pore water (ca. 0.05 $m^3 \cdot m^{-3}$) continues to exist even if the soil is completely frozen (Stein and Kane, 1983). The recorded anomalous zero values are probably related to the inadequate ability of the time domain reflectometry sensors to detect immobile residual liquid water. We believe that in these periods the simulation results appear more realistic.

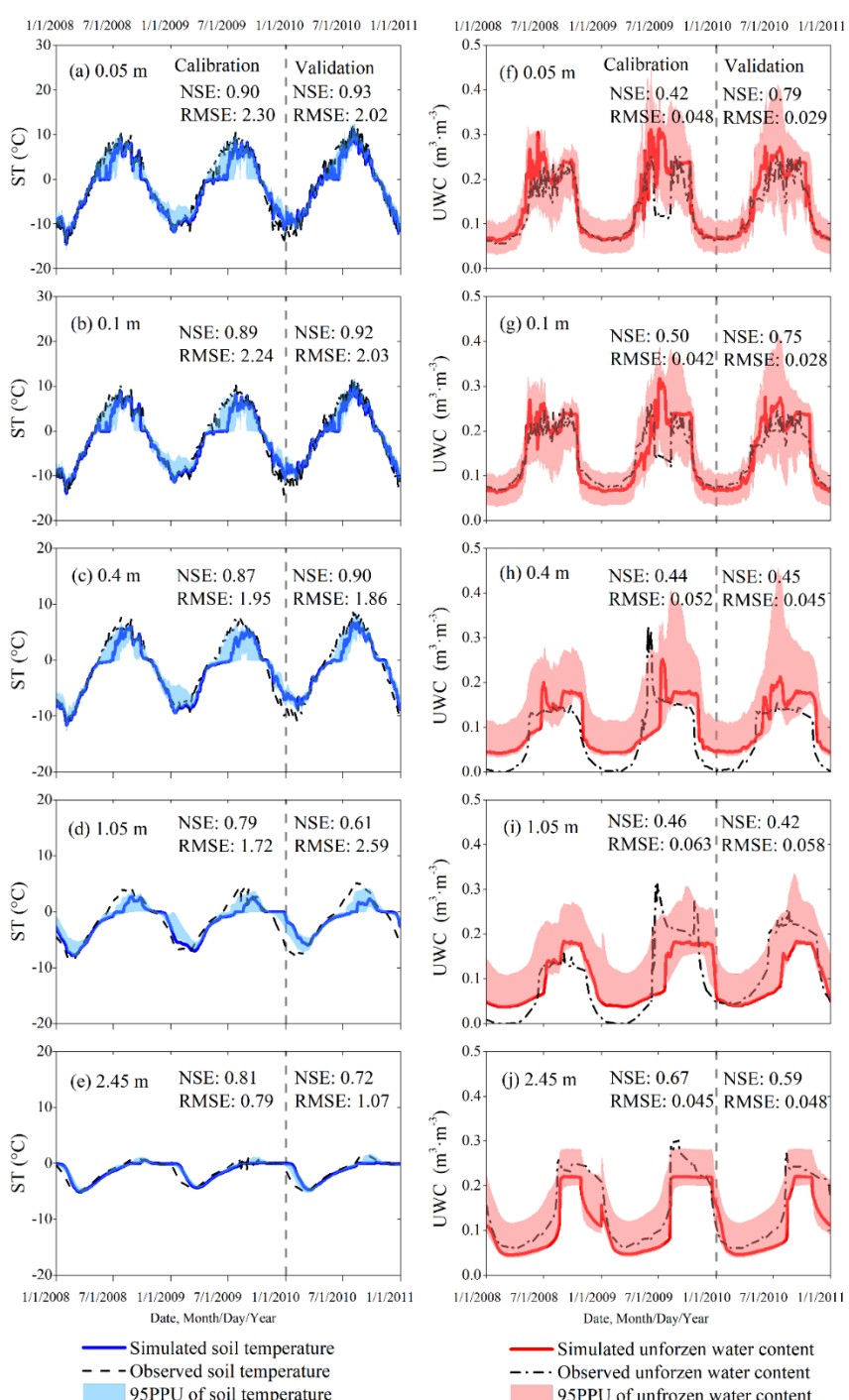

**Figure 2  Simulated (solid lines) and observed (dashed lines) daily soil temperatures (ST; left panels) and unfrozen water contents (UWC; right panels) at 0.05 (a and f), 0.1 (b and g), 0.4 (c and h), 1.05 (d and i) and 2.45 m (e and j) depths at the Tanggula (TGL) site from 1 January 2008 to 31 December 2010. The simulated soil temperatures (solid blue line) and UWCs (solid red line) are the results with the optimal parameter values identified from the 1000 random parameter combinations. NSE:  the Nash-Sutcliffe efficiency coefficient; RMSE: root mean square error. The 95PPUs of the model outputs are from all 1000 randomly generated parameter combinations.**

During the spring thaw period in each year, the observed temperatures (Figure 2a-e) increased rapidly from the negative to the positive, but the simulated soil temperatures exhibited an obvious prolonged duration of the zero-curtain effect, which delayed the warming of soil temperature by days. Accordingly, the 95PPUs of the simulated soil temperatures from 1000 random parameter combinations also exhibited larger intervals in spring thaw periods than in other seasons. This effect was especially strong in 2009. The formation of zero curtain is a joint result of multifaceted thermal processes including evapotranspiration, phase change, heat conduction and convection during freeze and thaw periods (Outcalt et al., 1990), and is more obvious during the thawing process than the freezing process (Jiang et al., 2018). The overestimation of the zero-curtain duration in the SHAW simulation is primarily related to the irrational vapor motion and simplified phase change between ice and liquid.

In January 2010, an overestimation of soil temperature was observed throughout the entire soil column (Figure 2a-e). However, this phenomenon did not occur in the same month in 2008 and 2009. It is certain that the observed discrepancies result from unusually warm air temperature (Figure 3). These anomalies also caused additional snowmelt events with ca. 0.5 mm of snow water equivalent in this month (Figure 3).

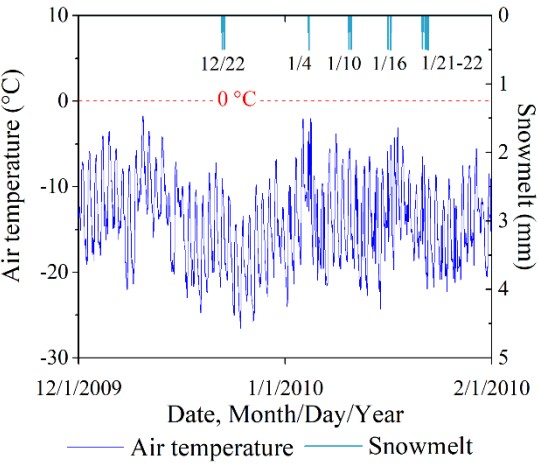

**Figure 3  Unusual warm air temperature in January 2010, presumably related to overestimation of modelled soil temperature that month, and simulated hourly snowmelt, also influenced by air temperature. The dates (month/day) at the top of the figure indicate when the snowmelt events occurred.**

**3.2 General characteristics of the convective heat transfer impacts**

The simulations of hourly soil temperature under the Control scenario using the original SHAW model with full CHT processes included are shown in Figure 4a. The differences in the soil temperature profiles

between the Control and the two other scenarios, i.e., partial (NoSurf) or full (NoConv) exclusion of CHT in the model, are presented in Figure 4b and Figure 4c, respectively, which depict the distribution patterns of CHT occurrence in time and depth, and the intensity of soil temperature variations due to CHT. The effects of CHT appeared primarily in the thaw periods of 2008 and 2010, as shown in Figure 4b and c, and resulted in a pronounced increase in soil temperature. However, during the same periods of 2009, no noticeable temperature differences were simulated between Control and NoSurf for the entire soil column, and only minor differences were simulated between Control and NoConv at intermediate depths. Since vapor convection was not altered, those effects were due entirely to the partial or complete presence of CHT due to surface infiltration and vertical advection within the soil column. The differences in soil temperature were noticeable even at shallow depths in January 2010 (Figure 4b and c), when soils at those depths were frozen and impermeable. This phenomenon coincided with the occurrence of extra snowmelt events during this period, as shown in Figure 3. Although snowmelt did not infiltrate into the underlying impermeable soil layers, it moved downward into the uppermost soil layer (0.00 m) during these periods. In the Control scenario, the sensible convective heat flux due to percolation of snowmelt water altered the temperature of the uppermost soil layer and consequently the temperature gradient there. These temperature perturbations were then transmitted to the near-surface soil layers by conduction. In the other two scenarios, where the CHT process at the ground surface were excluded from the model, the same amount of snowmelt water was transported to the top soil layer, but no convective heat was transferred and thus no thermal disturbance occurred in the shallow soil layers as in the Control, as manifested in the temperature deviations in the shallow layers when contrasting the scenarios. It suggests that CHT could also have indirect thermal impacts during freeze periods, providing that snowmelt occurs during these periods in response to changes in air temperature.

As shown in Figure 4b and c, the occurrence of CHT became increasingly delayed with increasing soil depth, with the largest delay occurring at the deepest location. The shallow depths are characterized with long thaw periods spanning from later spring to summer, with large thermal gradients and active water migration between soil layers, so that CHT at those depths has considerably impacts on the thermal regime. In contrast, the thermal effects of CHT are much smaller at deeper depths, where the temperature gradient and water motion are relatively modest. The differences between the Control and NoConv (Figure 4c) are more evident than those between the Control and NoSurf (Figure 4b) in particular at the

shallow and intermediate depths, suggesting that the CHT process within the soil also influences the soil thermal regime, although its effect is not as strong as that due to infiltration from the surface.

During spring thaw periods when air temperature is higher than ground temperature, snowmelt infiltrates and warms soils with warmer water temperature, as manifested by higher simulated soil temperatures under Control than under NoSurf or NoConv. However, Figure 4 also shows the moments when the simulated temperature is lower under the Control scenario than under NoSurf and NoConv, signifying the existence of a cooling effect of CHT, although this cooling effect is much weaker than the heating effect. The culprit is the direction of convective flux as well as the temperature difference between soil layers. When the flows move from a higher temperature layer to a lower temperature layer, the low-temperature layer is heated, and in the reverse direction, the high-temperature layer is cooled. It is interesting to note that compared to Control−NoSurf (Figure 4c), there are more negative differences (in blue) in Control−NoConv (Figure 4c). It implies liquid migration within the soil has more frequent cooling effects on the thermal regimes than surface infiltration.

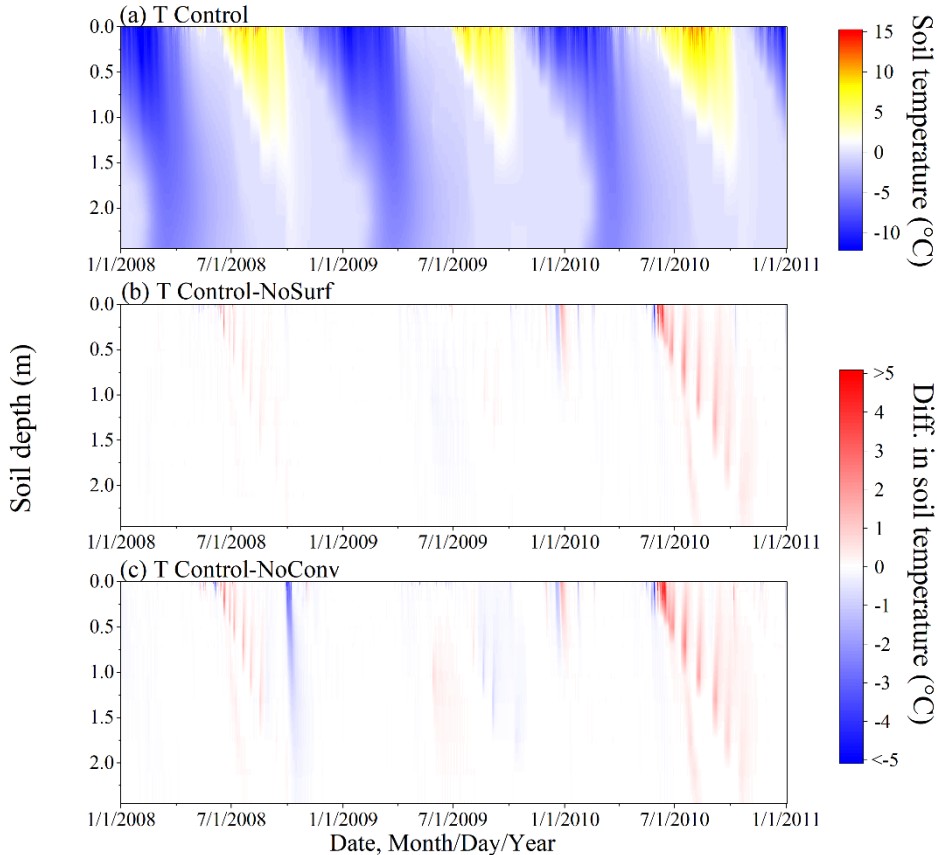

**Figure 4 Simulated hourly ground temperature profiles under the Control scenario (a), and the differences in soil temperature between the scenarios: Control−NoSurf (b) and Control−NoConv (c). Control, NoSurf and NoConv represent full, partial and no consideration of convective heat transfer (CHT) in the SHAW**

**model, respectively. NoSurf removes CHT due to infiltration and NoConv removes all CHT processes from**
**the model.**

The UWC differences between the scenarios (Figure 5) are comparable to soil temperature differences

in both space and time. The effects occurred mainly in thaw periods and were attenuated with increasing

depth. In late spring 2009, patterns of UWC differences differ markedly from those in the same months

of adjacent years (Figure 5b, c). The 2009 differences are confined to shallow depths, whereas in adjacent

445     years the differences penetrate most soil depths. The 2009 pattern of UWC also differ from the soil

temperature pattern in the same year (Figure 4b, c), where no temperature differences are observed at

shallow depths. This suggests that due to relatively less water migration during the thaw period, CHT

only promotes phase change and produces more liquid water from ice this year compared to

neighbouring years, but is unable to noticeably increase soil temperature.

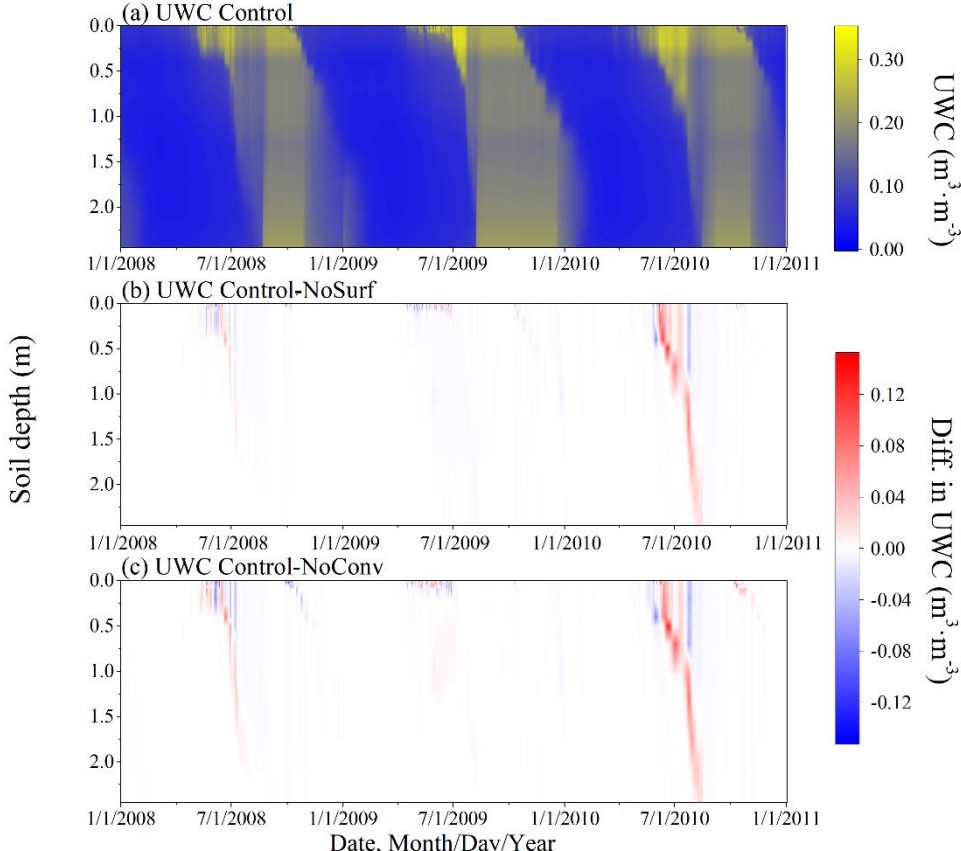

**Figure 5  Simulated hourly unfrozen water content (UWC) profiles under the Control scenario (a), and the**
**differences in UWC between the scenarios: Control−NoSurf (b) and Control−NoConv (c).**

### 3.3 Stratified effects of convective heat transfer

As shown in Figure 4a, CHT processes have very non-uniform thermal effects at different soil depths. Therefore, to illustrate the inconsistent impacts of CHT at depth and to identify the driving factors, three specific soil layers, i.e., the layer centered at 0.05 m depth closest to the ground surface and for which observations are available, the layer at 1.05 m depth, which is at the middle depth of the entire active layer, and the layer at 2.45 m adjacent to the bottom of the active layer, were selected to represent the shallow (0-0.2 m), intermediate (0.4-1.3 m) and deep depths (deeper than 1.75 m), respectively, where very different impacts of CHT were observed.

### 3.3.1 Shallow depths

By contrasting the hourly results at the shallow depths of NoSurf and NoConv with those of the Control, the effects of CHT were quantified, as shown in Figure 6 and Figure 7. The patterns are similar for all shallow depths (above 0.2 m). Generally, the effects of CHT on the thermal regime are strong at the shallow depths, as shown by the soil temperature differences at 0.05 m depth in 2008 and 2010 obtained from both Control−NoSurf and Control−NoConv (Figure 6a and Figure 7a). In the figures, positive differences in soil temperature represent the heating effects of CHT at the shallow depths, while negative values represent the cooling effects. The convective heat estimated in Control acts as an extra heat source during the spring melting periods when CHT occurred, the source that not only provided heat for the phase changes from ice to water, but also warmed the soils and caused an average increase in soil temperature of about 0.9 °C (shown in Table 2) , compared to the NoSurf and NoConv results. The maximum temperature warming could reach 10 °C at a certain time. In Figure 6 and Figure 7, the simulated soil temperatures under Control (black dash line in Figure 6c and Figure 7c) surpassed 0 °C during the final stage of melting, while the temperatures of NoSurf (blue line in Figure 6c) and NoConv (red line in Figure 7c) still remained at 0 °C. This indicates the ending of the zero-curtain effect was advanced by several days due to the heating effect of CHT as simulated in Control, compared to NoSurf with partial consideration of convective heat and NoConv without consideration of convective heat.

Apart from the imposed heating effect, an opposite cooling effect is observed at shallow depths, indicated as negative differences in Figure 6a and Figure 7a. Soil temperature decreased by an average of -0.79 °C and -1.06 °C (Table 2), respectively, when contrasting the results of Control with those of NoSurf and

NoConv during the spring thaw and fall freeze periods, with extreme temperature reduction by up to -5 °C occurring in some durations. The cooling effect is mainly related to the upward water flow induced by the hydraulic gradient during thaw periods and the negative temperature differences between the low-temperature surface and the high-temperature soils during infiltration.

Figure 4 already shows that more cooling events were triggered by convective processes within the soils than by infiltration. It becomes even clearer by comparing Figure 7a showing Control−NoConv with Figure 6a showing Control−NoSurf. Table 2 shows that there are 1757 cooling events in the comparison between Control and NoConv, but only 1195 between Control and NoSurf. Moreover, Figure 7a contains more nonzero values than Figure 6a. This is plausible because the results of Control−NoConv include

the entire impact of CHT, whereas Control−NoSurf includes only a portion of the surface infiltration. The peaks of liquid water migration in the 0.05 m layer are shown in Figure 6b and Figure 7b, where positive values indicate downward flows and negative for upward flows. The downward flows are related to snowmelt events as shown in Figure 6b and Figure 7b, where only those simulated under Control are shown because the snowmelt events under the three scenarios are nearly identical. It indicates that

infiltration of snowmelt is the major source of downward liquid flow during spring. Nevertheless, as the ambient temperature rises, the underlying frozen ground begins to thaw to depth (Figure 4b), and ground ice melt also partially contributes to the liquid flux. It is difficult to distinguish which fraction of the flux comes from snowmelt and which from ground ice melt. Thus, we used the total liquid flows instead of snowmelt volume to examine the relationship between soil water migration and CHT. Liquid water

migration correlates the occurrence of CHT very well (Figure 6a and Figure 7a). During the zero-curtain durations in 2008 and 2010, soils were repeatedly frozen and thawed. Liquid water migration became more frequent after soils were completely thawed and soil moisture content began to increase. At this time, CHT became more active at this depth. The situation was different in the spring of 2009, when a prolonged zero-curtain period was simulated and water flow in the soils was suppressed. As a result, only

marginal effects of CHT were observed during this period. In summer, when the zero-curtain had completely disappeared, the soils had a relatively stable soil moisture content. In this period, liquid water percolated mainly through the soils at a slow rate. The rate could sometimes reach half of the maximum liquid flux during the spring thaw. However, only a small

increase in soil temperature of about 0.1 to 0.5 °C, or approximately 10% of the spring warming, could

be attributed to the convective heat associated with water migration.

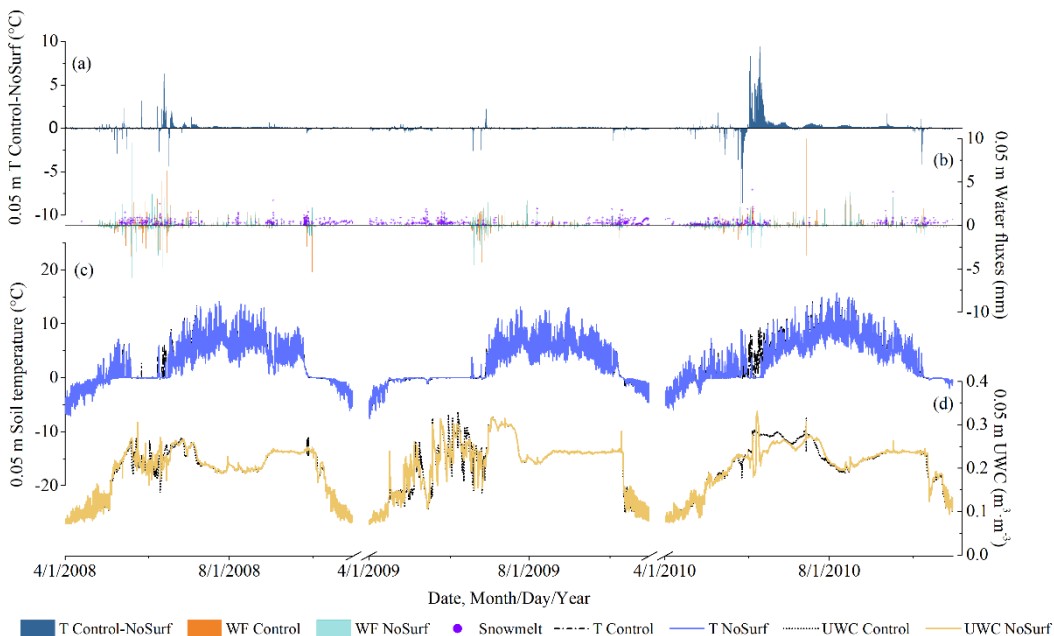

**Figure 6  Hourly soil temperature, water flux and UWC at 0.05 m depth, representative of shallow depths, simulated under NoSurf and Control during the 2008-2010 thaw periods. From top to bottom are: (a) the**

**differences in soil temperature (T) between Control and NoSurf (Control−NoSurf), with positive values indicating heating effects and negative values indicating cooling effects; (b) snowmelt water simulated under Control and the water fluxes (WF) at 0.05 m simulated under NoSurf and Control, where positive values represent downward flows and negative for upward flows; (c) soil temperatures and (d) UWCs simulated under NoSurf and Control.**

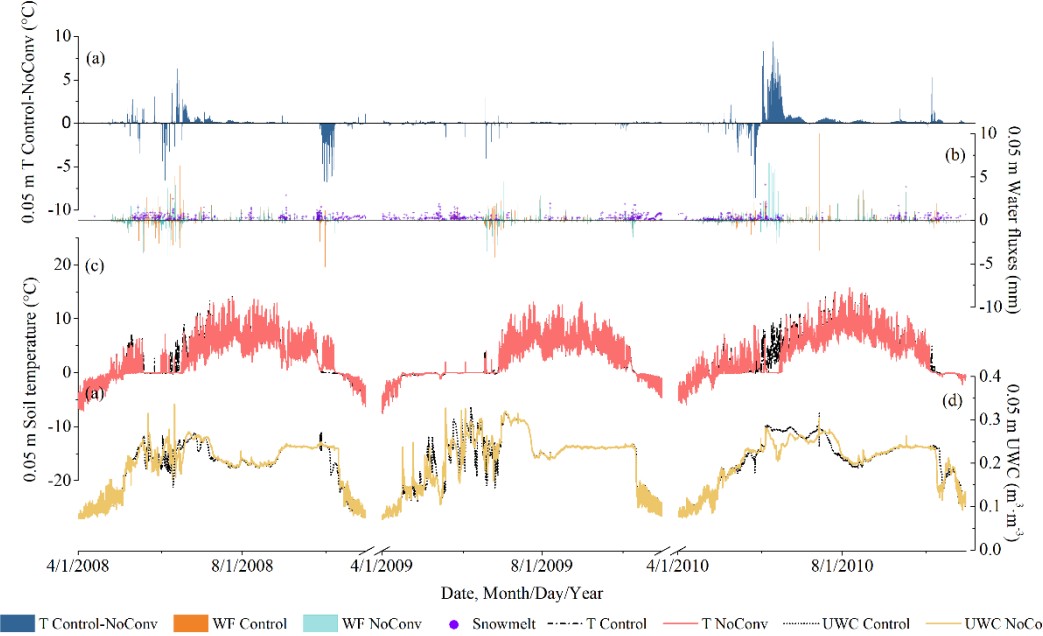


**Figure 7 Hourly soil temperature, water flux, and UWC at 0.05 m depth simulated under NoConv and Control during the 2008-2010 thaw periods. From top to bottom are: (a) the differences in soil temperature (T) between Control and NoConv (Control−NoConv), with positive values indicating heating effects and negative values indicating cooling effects; (b) snowmelt water simulated under Control and the water fluxes**

**(WF) simulated under NoConv and Control, where positive values represent downward flows and negative for upward flows; (c) soil temperatures and (d) UWCs simulated under NoConv and Control.**

**Table 2 The numbers of occurrences of heating and cooling CHT events and the average temperature deviations caused by CHT at 0.05, 1.05 and 2.45 m depths at TGL. The deviations of 0.1 °C or less were excluded for statistics.**

|  | Control−NoSurf | | | Control−NoConv | | |
|---|---|---|---|---|---|---|
|  | 0.05 m | 1.05 m | 2.45 m | 0.05 m | 1.05 m | 2.45 m |
| Number of heating events | 2436 | 1850 | 602 | 3109 | 2984 | 456 |
| Average increase (°C) | 0.86 | 0.43 | 0.21 | 0.97 | 0.41 | 0.23 |
| Number of cooling events | 1195 | 189 | 10 | 1757 | 1302 | 67 |
| Average decrease (°C) | -0.79 | -0.24 | -0.20 | -1.06 | -0.41 | -0.20 |

**3.3.2 Intermediate depths**

In contrast to the strong effects at shallow depths, the effects of CHT on thermal and hydrological regimes are not as pronounced at the intermediate depths of the active layer, as can be seen in Figure 8 and Figure 9, which show the hourly results at 1.05 m depth, because water migration was inactive at these depths. However, the characteristics of the occurrences were similar to those at the shallow depths, despite the

weaker values. Temperatures at these depths averaged 0.43 °C(Table 2)higher in Control than in

NoSurf during thaw periods when CHT events occurred. The comparison also shows that the complete

thaw in the active layer occurred about one day earlier in Control, which is due to the heating effect of

convective heat penetrating downwards from the surface that is present in Control. The cooling effect of

convective heat was also observed within the soil layers during certain periods. In these cases,

temperatures in NoConv surpassed those in Control by an average of 0.41 °C for each cooling event

(Table 2), while temperatures in NoSurf surpassed those in Control by only an average of 0.24 °C (Table

2). The frequency of cooling CHT effects when comparing Control and NoConv (1302 times over the

entire simulation period) was also several times higher than when comparing Control and NoSurf (189

times), indicating that liquid flux between soil layers exerts more cooling effects on soil temperature at

intermediate depths than infiltrative flux does. This is primarily attributed to the joint effect of weak

infiltration from the surface and upward water fluxes within the intermediate depths, which carry cooler

water from the lower depths to the upper depths.

Another notable dissimilarity to shallow depths is the apparent incongruity between the occurrence of

CHT and peak water migration at intermediate depths. When vertical advection occurs at the intermediate

depths, the small amount of heat along with advection can hardly satisfy the consumption of ongoing

phase change, which requires a large amount of heat, and thus it is usually not possible to directly increase

soil temperature of the lower layer. However, this process alters the thermal gradients in the soil column,

which gradually affects the total thermal regime. This is a delayed and slow response that results in

asynchronous spikes in temperature and water fluxes, as shown in Figure 8 and Figure 9.

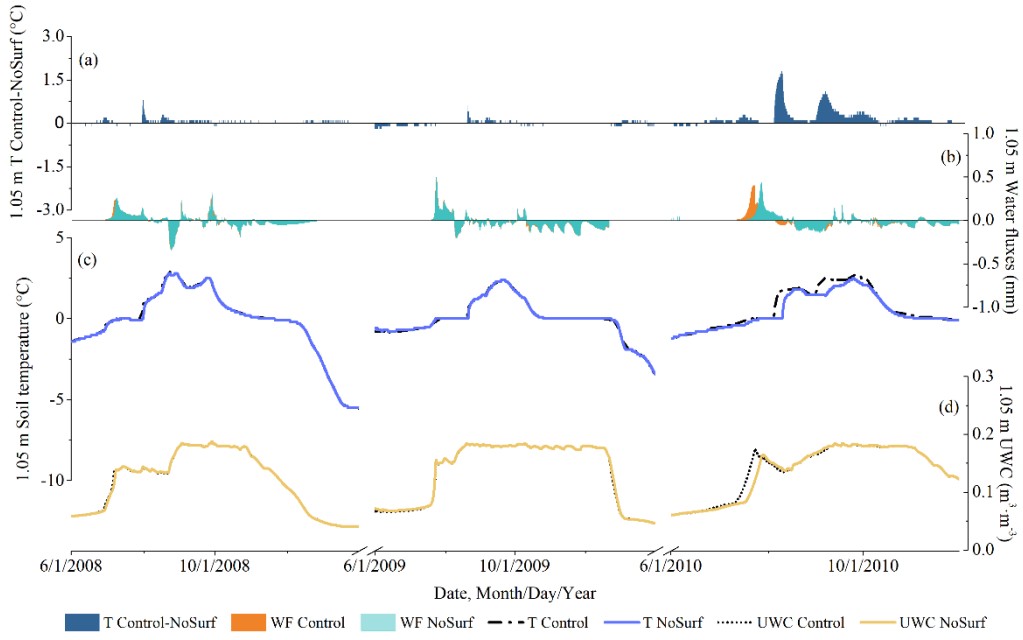


**Figure 8  Hourly soil temperature, water flux and UWC at 1.05 m depth, representative of intermediate depths, simulated under NoSurf and Control during the 2008-2010 thaw periods. The same notations as in Figure 6 are applied.**

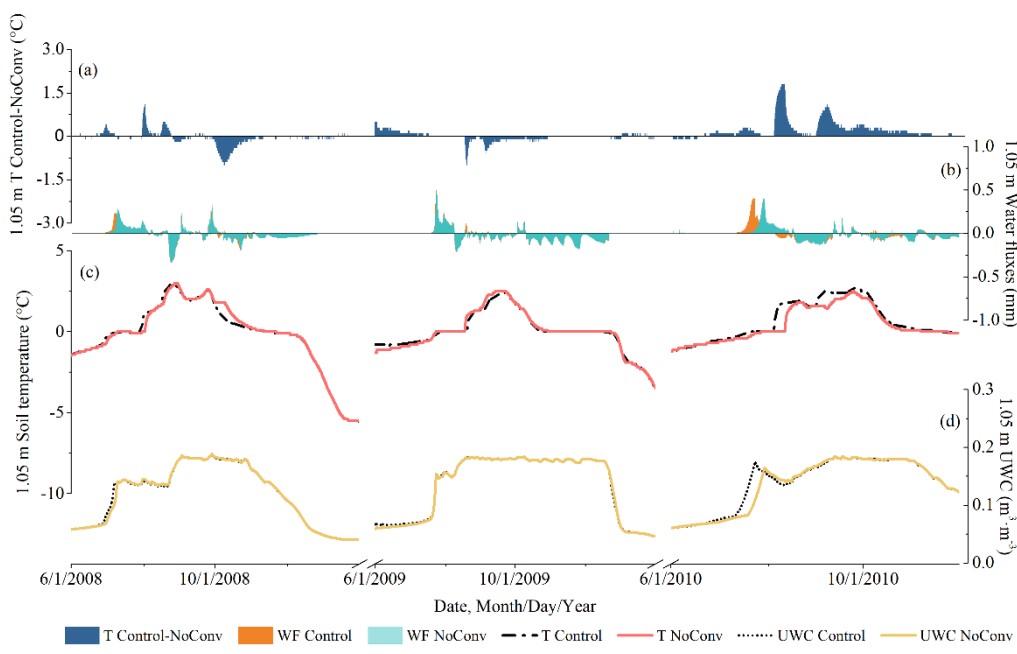

**Figure 9  Hourly soil temperature, water flux and UWC at 1.05 m depth simulated under NoConv and Control during the 2008-2010 thaw periods. The same notations as in Figure 7 are applied.**

### 3.3.3 Deep depths

The thermal impacts of CHT were minimal at the deep depths, as shown in Figure 10a, Figure 11a and
Table 2. Accordingly, water flow occurred infrequently at a depth of 2.45 m near the bottom boundary
of active layer (Figure 10b and Figure 11b), with a much lower frequency than at the shallow and
intermediate depths. Soil temperature remained at zero degrees during many thaw periods (Figure 10c
and Figure 11c). According to the study of Romanovsky and Osterkamp (2000), the CHT associated with
advection of unfrozen pore water no longer affects soil temperature when the ambient soils hold a
temperature close to the frozen point that is same as the migrating liquid. The presence of temperature
gradient between depths is a prerequisite for the thermal impacts of CHT. At the deep depths, however,
the soil temperature varies only slightly over the course of a year and differs in little from that of advective,
unfrozen water. Therefore, some marginal temperature differences (about 0.2 °C on average, as shown
in Table 2) were observed in this study when comparing NoSurf/NoConv with the Control. It indicates
that the thermal effects of CHT are marginal at the deep depths of the active layer and can usually be
ignored, although vertical advection processes may occasionally be observed.

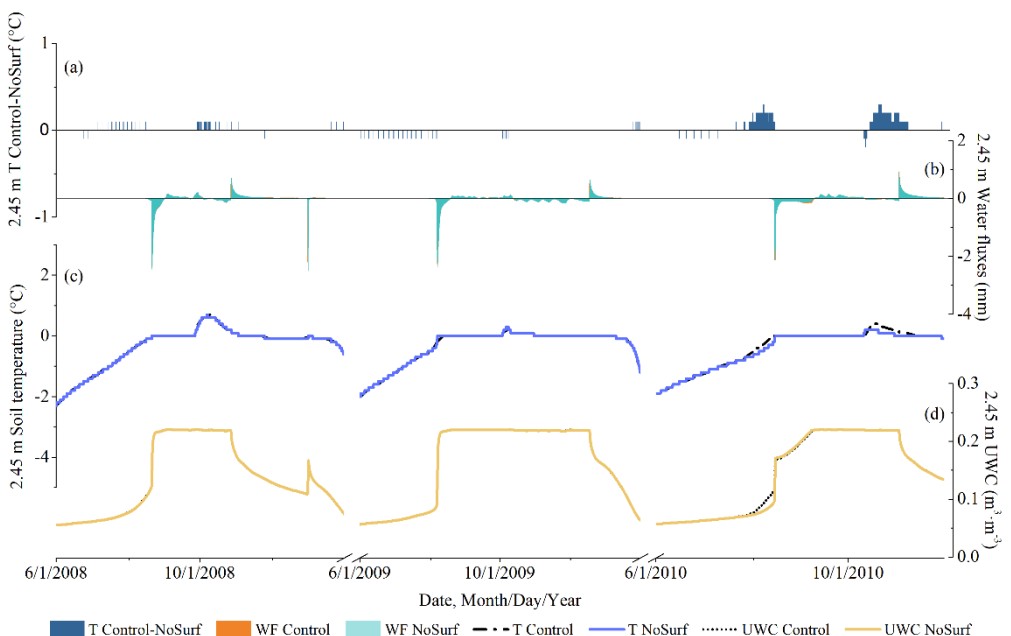

**Figure 10  Hourly soil temperature, water flux and UWC at 2.45 m depth, representative of the deep depths,
simulated under NoSurf and Control during the 2008-2010 thaw periods. The same notations as in Figure 6
are applied.**

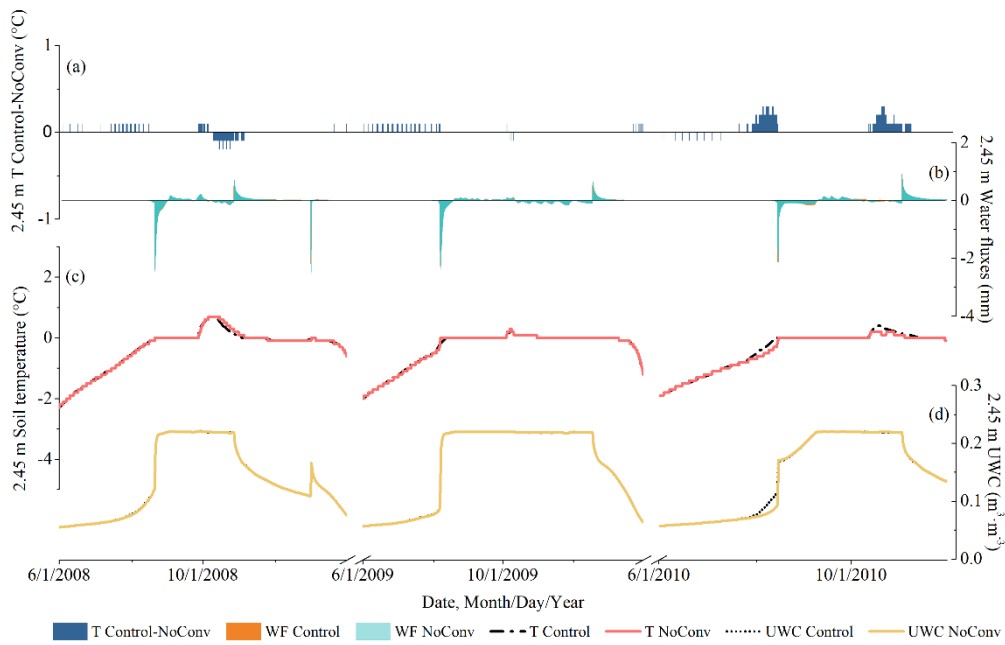

**Figure 11  Hourly soil temperature, water flux and UWC at 2.45 m depth simulated under NoConv and Control during the 2008-2010 thaw periods. The same notations as in Figure 7 are applied.**

## 4.    Discussion

**4.1 Bidirectional thermal impacts of convective heat transfer**

This study has investigated two types of liquid CHT, i.e., the one due to infiltration from surface snow melt or rainfall and the other occurring within the soil column due to hydraulic gradient, using a numerical modelling approach. During thaw periods, soil temperature generally declines from the surface downward and toward depth. Thus, infiltrative water moves downward at warmer temperatures and exerts a heating effect on the soils passing through, likely increasing soil temperature and accelerating the process of phase change, especially in the later spring. Previous studies have also observed some CHT effects due to the anomalous fluctuations in soil temperature during the periods when snowmelt or rainfall infiltrates into the soil. Kane et al. (2001) and Hinkel et al. (1996) reported the step-like increase in near surface temperature by 2 °C to 4°C and the pronounced disruption of thermal gradients during the periods when snowmelt infiltrates. The increase in soil temperature (Iijima et al., 2010; Mekonnen et al., 2021) and frost front depth (Douglas et al., 2020; Guan et al., 2010) was also observed following heavy rainfall events, indicating that precipitation is another important source of CHT in addition to

snowmelt. Kane et al. (2001) estimated that the CHT during heavy precipitation is twice as high as conductive heat. Hinkel et al (2001) measured 0.5 °C and 1.3 °C positive changes in soil temperature in response to infiltration from snowmelt and rainfall, respectively. Although the maximum temperature perturbation due to the CHT process could occasionally reach 10 °C in our study, most of the temperature perturbations were limited to less than 5 °C, which is consistent with the observed phenomenon. Although these observed warming events are mainly driven by liquid convection, they are still a combined product of multiple heat transfer processes, including conduction and vapor flow. On the other hand, our modelling study provides a good explanation for the mechanism behind these observations, how those warming events at certain depths during spring thaws and rainfall events could occur due to CHT processes alone.

However, CHT also likely has a cooling effect within the active layer. The actual role of CHT at a given time depends on the direction of the liquid flow and the temperature difference along the flow path at that time. When the air temperature and surface temperature drop rapidly below the subsurface soil temperature, water flow from the ground surface into the soil layer can reduce the soil temperature, as demonstrated in our contrasting experiments where the soil temperature in Control, which fully accounts for CHT, was lower than that in NoSurf, which ignores infiltrative convective heat, at the shallow depths (0-0.2 m) in some time periods. The other cause of cooling is related to upward water migration, such as the return flow simulated in the SHAW model during the thaw period when the lower depth is colder than the upper depth in the ground. By contrasting Control−NoSurf with Control−NoConv, we found many cooling events occurring at intermediate depths (0.4-1.3 m) within the soils and associated with upward water migration driven by the hydraulic gradient, resulting in higher simulated soil temperature simulated in NoConv (which completely removes the CHT process) than in Control. Some previous studies (Gao et al., 2020; Li et al., 2016) have reported that the melting occurring at the permafrost table provides water supply to the upper depths. Effects on the thermal and hydrological regimes of the entire active layer due to the upward liquid movement have also been reported (Chen et al., 2019; Cui et al., 2020; Rowland et al., 2011). Our study strengthens those existing studies by quantifying and interpreting such effects from a modelling perspective.

To ensure that the experiment at TGL was not an exception, we performed a similar simulation (spanning 2015-2018) at another permafrost site (coded QT08) near the town of Wudaoliang alongside the Qinghai-

Tibet highway to verify the bidirectional thermal impacts of CHT. Since hourly meteorological observations are not available for QT08, we extracted the atmospheric forcing at the corresponding point from a reanalysis dataset with a resolution of 0.1° and 3 h resolution (He et al., 2020). This dataset has been widely used for permafrost simulations on the QTP due to its high accuracy in this region (Wu et al., 2018; Yang et al., 2021; Zhang, G. et al., 2021). The bidirectional thermal impacts of CHT were well observed at QT08 although the model was not well informed owing to, for example, coarse-resolution forcing data and uncalibrated parametric values for all soil layers. The role of CHT at QT08 was primarily to warm the ground during the spring thaw, along with a small number of cooling effects (Figure A1 and Table A1). Compared to the TGL results, the magnitude of CHT-induced soil temperature fluctuations at QT08 was attenuated by a longer 3 h time step and was not as pronounced as at TGL (Figure A1 and Table A1) because CHT is the rapid heat exchange process that normally occurs in a hourly or less interval. Although the accuracy of the QT08 simulation was not as good as that of TGL (Figure A2), this experiment provides useful evidence that the bidirectional thermal effects of CHT can be common in alpine permafrost environments during the thaw period. Appendix A gives more details regarding the simulation at QT08.

Summer rainfall was believed to play an important role in modulating the thermal regime in the active layer (Wright et al., 2009; Zhang, G. et al., 2021). While Rachlewicz and Szczuciński (2008) postulated that non-conductive heat due to rainwater infiltration was particularly important for the thermal regime in the uppermost soil at about 5 cm depth, Kane et al. (2001) estimated that the CHT due to heavy precipitation likely doubled conductive heat. However, this study shows the effects of CHT due to summer rainfall were much less than during spring thaw periods (Figure 6a). In Figure 6b, downward water fluxes (shown as positive values) respond well to summer rainfall in the near surface ground, while the impacts on soil temperature (Figure 6a) at this depth due to CHT are minimal. Those findings are not in conflict. Summer precipitation has multifaceted non-conductive effects, including cooling the topsoil through enhanced evapotranspiration from the ground surface, modifying soil properties such as heat capacity and conductivity by adding more liquid water to the soil (Zhou et al., 2021), rapidly transporting external heat to the soil through percolation, and providing heat for the melting process occurring at the freeze-thaw front as a heat source when additional liquid water accumulates above the front (Zhang, G. et al., 2021). In this study, hydraulic and hydrological functions of precipitation are the same in the

scenarios, therefore, only the effect of rapid transportation of external heat into the soils is associated with the CHT process under investigation, which was found to be of less importance among the diverse effects of summer precipitation. The same weak significance of CHT associated with summer rainfall is also reported by Zhang, M. et al. (2021), where the positive energy flux of rainfall convection into soil layers was low, in contrast to other negative energy fluxes due to increased soil evaporation and latent heat, and decreased soil conductivity due to growing soil moisture.

**4.2 Snowmelt influence on convective heat transfer**

Snowmelt is a main component of spring infiltration that causes CHT from the ground surface and warms the underlying soil layers, as evidenced by both our study and the observation-based studies mentioned above. Normally, as air temperature warms, snowmelt is accompanied by thawing of the soil, allowing meltwater to freely infiltrate into the soil. This is the main form of snowmelt-induced CHT. However, as shown in our contrasting experiments, the soil temperature of the Control scenario differs from that of the NoSurf and NoConv scenarios even during the periods when no vertical water flux occurs in these layers. This suggests that snowmelt could also have an indirect influence on the soil without explicit infiltration. For example, some snowmelt events took place in January 2010 (Figure 3), possibly due to a transient increase in air temperature or shortwave radiation, resulting in temperature differences between Control and NoSurf/NoConv at shallow depths (Figure 4). However, snowmelt could not infiltrate because the ground was still in an impermeable, frozen state at this time. We assume that in this case the temperature at the ground surface was affected by the convective heat carried with snowmelt, although snowmelt can only move downward through the snowpack and reach the ground surface before it drains laterally, which then affects the thermal gradient at shallow depths. Thus, the altered temperature at the ground surface was spread to the lower layers by the changed thermal gradient. Not coincidentally, we also measured some CHT-induced soil temperature changes at intermediate depths, but at the same time no corresponding convective fluxes were observed, which we believe is also part of the indirect convective heat influence exerted from other depths to this depth by conduction. Although this type of heat transfer is accomplished through heat conduction, it is still essentially an indirect convection-induced heat transport.

Moreover, percolation of liquid water within snow leads to a complex spatial redistribution of snow depths and densities, which strongly regulates the ground temperature and the active layer thickness in snowpack areas (Magnin et al., 2017). Zweigel et al.(2021) have reported that redistribution of snow, taking into accounts snow water percolation, increased ground surface temperature 1-2 °C, demonstrating another aspect of the indirect impacts of snow water migration on the permafrost thermal regime.

## 4.3 Effects of soil moisture migration in late spring

In permafrost regions, soil moisture migration within the active layer is a major form to support CHT. Liquid water migration at shallow depths occurs most frequently during spring thaw periods, transporting considerable heat into soils and producing notable thermal effects on soil temperature parallel to these water migration events. Measurements of UWC at some typical permafrost sites indicate that UWC rises rapidly to the highest in late spring as the ground ice in the active layer melts, before gradually falling back to field capacity by summer (Boike et al., 1998). Before the thaw begins, excessive ground ice is present in the shallow layers because soil permeability is reduced during the freezing process, preventing the upper liquid water from percolating to depth, and because a potential gradient exists between the constantly downward migrating freezing front and the underlying unfrozen layer. The segregation potential of frozen ground drives the liquid flux upward to the front. As a consequence, the frozen shallow layers tend to hold excessive ice content much more than the liquid equivalent can be held (i.e. field capability) in the melting soil (Cheng, 1983; Perfect and Williams, 1980). SHAW accounts for the decrease in permeability due to growth of ground ice, but ignores the mechanism of segregated ice. Despite this deficiency, nearly saturated liquid water can be simulated at the onset of the thaw, and the fraction in excess of field capacity moves downward or upward as return flow. This explains the emergence of the frequent and intense water migration in late spring, which makes strong CHT possible at those specific depths and times.

In addition, Kurylyk et al. (2016) mentioned the potential thermal impacts caused by lateral discharges in permafrost regions, while relatively small. Unfortunately, it is not investigated in this study because the one-dimensional SHAW model ignores lateral water migration from the perimeter into the soil column due to soil anisotropy, which may lead to some uncertainty in simulating water flow within the active layer. Recently, 3-D LSMs have emerged as found in the recent literature (Endrizzi et al., 2014;

Painter et al., 2016; Rogger et al., 2017). Those models couple both horizontal and vertical thermal and hydrological processes at the surface/subsurface and own obvious advantages over 1-D models in terms of the physical basis for studying surface/subsurface water migrations and the thermal consequences on permafrost. Although current observations on the QTP cannot meet high data and parameter demands of 3-D models for, this provides another direction for CHT studies on the QTP.

## 4.4 Limitations

CHT is a rapid thermal exchange process that generally occurs at hourly intervals and is influenced by soil moisture, so hourly data of high quality are used for CHT study. However, due to the harsh natural environment and cumbersome transport, we are unable to obtain a large amount of long-term, high-quality climate and permafrost observation data from multiple sites. Though we have conducted a three-year long-term experiment at the TGL site that included two years of significant CHT impacts (2008 and 2010) and one year of relatively weak impacts, which could support investigation of interannual differences of CHT and the influencing factors associated, we still face the lack of high-quality data to enable a spatial and longer investigation into the CHT impacts. The good news is that some new observation sites have been deployed during the ongoing campaign of the second Tibet expedition (Chen et al., 2021), and the data situation for CHT studies is improved.

Existing observation-based studies indicate that the unfrozen pore water can still migrate under the capillary force and van Der Waals force, even when the ground is completely frozen (Fisher et al., 2020; Kane and Stein, 1983). The SHAW model is theoretically unable to simulate this process during the completely frozen periods. It also assumes that the direction of vapor flux is the same as that of the liquid water and adopts a simplified formulation of air flow in soils. This may bias the calculation of vapor convection. According to previous studies (Li et al., 2010; Yu et al., 2020), evapotranspiration together with CHT due to vapor flow plays an important role in near surface soils in thaw periods. Apart from the convective effect, air fluxes passing through soils may also alter thermal properties such as freezing point (Ming et al., 2020) or infiltration rate (Prunty and Bell, 2016). Such oversimplified assumptions in SHAW are possible factors contributing to a prolonged zero-curtain period in the simulation. In addition, SHAW permits the long-term coexistence of mixed solid-liquid state in physics. In reality, it is hard to maintain a long-term coexistence due to the inhomogeneity of soil properties and the interference of

environmental conditions (Akyurt et al., 2002). In this study, we subtract the results of the two scenarios with modified models from those of the control scenario with the original SHAW model, the uncertainties associated with these weaknesses are largely reduced, and the results are thus more reliable.

The SHAW model implements a special Newton-Raphson procedure to solve energy and mass balance equations, in which automated division into finer time steps occurs if the solution is not satisfactory. In this process of iterating over finer time steps, high quality upper and lower boundaries are necessary to maintain high simulation accuracy (Chen et al., 2019; Flerchinger, 1991). However, a byproduct of this process is that the importance of heat conduction arising from the boundaries is amplified as iterations proceed and as a consequence, the effects of nonconductive heat transfer are likely to be underestimated. Therefore, in the TGL application, the inaccurate lower boundary conditions for the SHAW model, particularly soil moisture, which is subject to appreciable measurable errors, also adversely affected the accuracy of the simulation at depth.

## 5. Conclusions

In this study, the SHAW model was applied to a typical permafrost area, the Tanggula site on the eastern Qinghai-Tibet Plateau, to explore and quantify the effects of liquid CHT on the active layer thermal regime. By modifying the SHAW model, we conducted a control experiment consisting of three scenarios representing the cases with full, partial or no consideration of CHT in the SHAW model. The following conclusions were drawn:

(1) The SHAW model performed well in simulating soil temperature and moisture dynamics in the active layer. The NSE values for the simulated temperature and moisture content in most soil layers exceed 0.70 and 0.45 in both calibration and validation periods, respectively.

(2) Liquid CHF is most likely to occur on the QTP in later spring and summer when frozen ground is fully thawed at shallow (0-0.2 m) and intermediate (0.4-1.3 m) depths. The infiltrative snowmelt and precipitation from the ground surface into the active layer is the main form of CHT in permafrost regions. Only minimal influences of convective heat were observed during freeze periods, due to some incidental snowmelt events in winter.

(3) At shallow depths (0.0 m to 0.4 m), CHT is more active during spring thaw period than in summer. During the spring thaw period, the differences in soil temperature simulated with or without considering

CHT had a wide range from -5 to 10 °C, whereas in summer the differences were about 0.5 °C, 10% of the value in spring, although the peak convection flux is comparable to that in spring. In the intermediate layer (0.4-1.3 m), much smaller impacts of CHT and no obvious seasonal variations were simulated due to the weakened convective flow, so that the ground temperature changed by only about -1.0 to 1.5 °C. The thermal effects of CHT are minimal at the deep depths of the active layer and can usually be ignored. (4) CHT has proven bidirectional thermal impacts on the active layer temperature, although the heating effect predominates in an annual freeze-thaw cycle on the QTP. During spring thaw periods, it was simulated that the soil temperature was averagely about 0.9 °C higher at shallow depths and 0.4 °C higher at intermediate depths when infiltrative convective heat considered was considered than no infiltrative heat transfer was considered, and the closing dates of the zero curtain were considerably advanced. Meanwhile, the opposite cooling effect due to upwelling liquid fluxes and thermal differences between soil layers can lower the simulated soil temperature by about -1 °C at shallow depths during some periods of spring, as indicated by comparing the simulation with CHT considered to the simulation without CHT. By contrasting the simulation ignoring CHT due to infiltration with the simulation completely ignoring CHT, the liquid CHT processes within the soils at intermediate depths led to a more significant reduction of about -0.4 °C on average in temperature and to several times higher frequency.

**Appendix A: SHAW simulation at the Wudaoliang site (QT08)**

Site QT08 was set up since 2009 to monitor water and heat dynamics in the active layer and is located at 35.22° N and 93.08° E near the town of Wudaoliang alongside the Qinghai-Tibet highway (Figure 1b). It is covered by alpine desert steppe. Permafrost is beneath 2.4 m. A nearby meteorological station in the Wudaoliang town can provide daily observations of atmospheric elements. The site provides daily observations of soil temperature and moisture content at 0.1, 0.4, 1.2, 2.0 and 2.4 m depth from January 1, 2012 to December 31, 2018 for this study. However, numerous zeros and missing values exist in the soil moisture data, preventing us from using it for validation. In general, this is not an ideal site for the SHAW model to precisely simulate the occurrence of convective heat transfer (CHT). CHT often occurs in a short interval, so the model must be fed hourly or higher resolution forcing data and observations that are not available at this site. The purpose of this simulation is to ensure our finding on the

bidirectional thermal impacts of liquid CHT was not an exception at TGL and can be reproduced at other

permafrost sites.

Based on the borehole information, the soil column was stratified into 9 layers (centered at 0.00, 0.02, 0.07, 0.13, 0.23, 0.39, 0.66, 1.11 and 1.84 m) with discrete soil texture types. The values for the vegetation and soil parameters were determined by the lookup table provided with the Noah land surface model as in our previous work (Wu et al., 2018). For the remaining parameters, the default values of the

model were used. Since hourly meteorological data are not available for this site, we extracted the meteorological forcing at the corresponding point from the gridded China Meteorological Forcing Dataset (CMFD) with a resolution of 0.1°and 3 h (He et al., 2020). The CMFD has been widely used for regional permafrost modelling on the QTP (Wu et al., 2018; Yang et al., 2021; Zhang, G. et al., 2021) and has good accuracy, especially on the eastern QTP, because most of its data were collected on the

eastern QTP. The lower boundary was set at 2.4 m depth based on daily observations of soil temperature and moisture at this depth. The simulation period was from January 1, 2015 to December 31, 2018.

The Control and NoConv scenarios were simulated at QT08 using the original SHAW model and the modified model (with CHT removed), respectively. The occurrences of heating and cooling events of CHT were counted and analysed following Eq. 12.

The simulation results at QT08 clearly reflect the bidirectional thermal impacts of CHT, although the model was not carefully configured, for example, as it used coarse-resolution forcing data and uncalibrated parametric values for all soil layers. The role of CHT at QT08 was primarily to warm the ground during the spring thaw period, along with a small number of cooling effects (Figure A1). The magnitude of CHT-induced soil temperature deviations at QT08 (mean about 0.2 °C, range -4 to 4 °C)

was not as pronounced as at TGL site (mean 0.9 °C, range -5 to 10 °C, Figure 4) because the longer modelling time step of 3 h attenuated the thermal influence of CHT, which is a rapid heat exchange process that normally occurs in an hourly or less interval. There was more active heat exchange and water migration in the deep layer (1.84 m, Table A1) at QT08 than in the deep layer (2.45 m, Table 2) at TGL, resulting in more frequent cooling events occurring in the deep layer at QT08. Both layers are near the

bottom of the active layer, but the thickness of the active layer at QT08 is much less than that at TGL. Due to the coarse spatial resolution of the forcing data (a grid of 0.1°), model performance for QT08 (Figure A2) was not so good as for TGL, which was driven by in-situ meteorological observations. At

QT08, soil temperatures at depth were generally overestimated. This experiment provides useful
evidence that the bidirectional thermal effects of CHT may be common in alpine permafrost

environments during thaw.

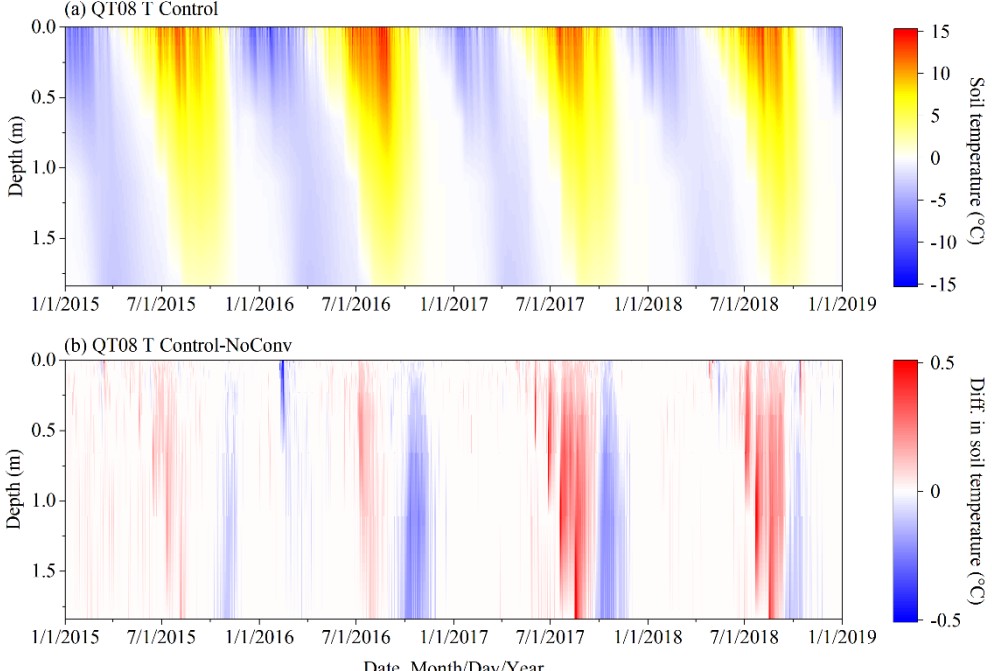

**Figure A1 Simulated ground temperature profiles at QT08 under the Control scenario (a), and the differences in simulated soil temperature between the Control and NoConv scenarios (Control−NoConv) (b). Control and**
**NoConv represent the scenarios with full and no consideration of convective heat transfer (CHT) in the SHAW model, respectively. Note, this figure has a different scale than Figure 4.**

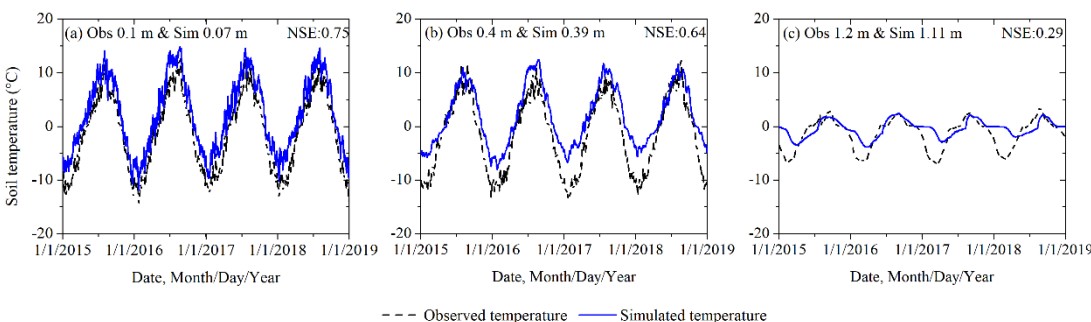

**Figure A2 Validation of simulation results at different depths at QT08 against the observed daily soil temperatures during January 1, 2015 to December 31, 2018. (a) for simulation depth of 0.07 m and**
**observation depth of 0.1 m; (b) 0.39 m for simulation and 0.4 m for observations; and (c) 1.11 m for simulation and 1.2 m for observations.**

**Table A1 The numbers of occurrences of heating and cooling CHT events and the average temperature deviations caused by CHT at 0.07, 0.66 and 1.84 m depths at the QT08 site. The deviations of 0.1 °C or less were excluded for statistics.**

| | Control−NoConv | | |
| --- | --- | --- | --- |
| | 0.07 m | 0.66 m | 1.84 m |
| Number of heating events | 405 | 1250 | 241 |
| Average increase (°C) | 0.16 | 0.20 | 0.31 |
| Number of cooling events | 149 | 288 | 348 |
| Average decrease (°C) | -0.27 | -0.17 | -0.20 |

## Code and data availability

The source codes of Simultaneous Heat and Water (SHAW) Model are available on the USDA Agricultural Research Service website (https://www.ars.usda.gov/pacific-west-area/boise-id/northwest-watershed-research-center/docs/shaw-model/, accessed November 29, 2021). The meteorological driving data and measured temperature and moisture data of the active layer at the Tanggula site under study can be downloaded from National Cryosphere Desert Data Center (https://www.ncdc.ac.cn/portal/, accessed November 29, 2021). The China Meteorological Forcing Dataset, which was used as the driving force in the QT08 simulation, can be obtained from National Tibetan Plateau Data Center (https://data.tpdc.ac.cn/, accessed January 24, 2022).The modified codes and simulation results of this study are openly available at https://doi.org/10.6084/m9.figshare.14827959.

## Author contributions

Z.N. and L.Z. conceived and conceptualized the idea; Y.Z. and Z.N. developed the methodology; Y.Z, Z.N. and H.J. performed the analyses; Z.N. and L.Z. acquired the funding and provided the resources; Z.N. supervised the study; Y. Z. wrote the draft version; Z.N., Y. Z., and H.J. reviewed and edited the writing.

## Competing interests

The authors declare that there is no conflict of interest.

**Acknowledgements**

This study was supported by National Natural Science Foundation of China grants 41931180 and 41971074. The authors thank the National Cryosphere and Desert Data Center for providing data and the USDA Agricultural Research Service for providing the model. The authors are especially grateful to Prof. Tonghua Wu and Jianzong Shi from the Cryosphere Research Station on the Qinghai-Tibet Plateau, CAS, for providing observed soil temperature and moisture content data at the Tanggula site and the QT08 site.

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
