# Peer review of "Convective Heat Transfer of Spring Meltwater Accelerates Active Layer Phase Change in Tibet Permafrost Areas"

_The Cryosphere, 2021_

## Author Comment (AC1)

**Response to Reviewer #1:**

We would like to thank Reviewer #1 for these valuable comments. Below are the responses to all Reviewer #1's comments and how we address them in the revised manuscript. **Texts in red** are the reviewer's comments; **those in black** are the authors' explanations to the reviewer's comments; and *those in blue* are the revised texts in the revised manuscript.

**RC1:** Convective Heat Transfer of the Spring Meltwater Accelerates Active Layer Phase Change in Tibetan Permafrost Areas
By Zhao et al.,
In this study, the authors applied the SHAW model to investigate temperature dynamics at the Tanggula site at the Eastern Qinghai-Tibetan Plateau. The modeling experiment includes the cases: 1. control (fully coupled diffusion convection), 2. NoSurf (no convective heat flux between surface and subsurface, but preserves the convective flux in the subsurface as a results of water movement in soil pores), 3. NoConv (no convective heat flux from surface to subsurface as well as no flux due to water movement in the subsurface). If I understand correctly, in case 2, the second left term in eqn. 2 remains for the subsurface formulation and is completely removed from the eqn. 2 for case 3.
Overall, the study is interesting and well presented.
**Authors:** Thank you for your positive comment.

**RC1:** I suggest including a discussion on a similar mathematical formulation described by Painter et al., (2016). How is the current mathematical formulation different from Painter et al. What are pros and cons of each formulation?
**Authors:** Thank you very much for providing this information. We have not previously mention Painter's work because it is related to a 3-D model that is not applicable to the study area (Qinghai-Tibet Plateau, QTP) due to the very high data requirements. We used a 1-D model instead to perform the permafrost simulation. We have read his paper and other related papers during this revision period, and found them to be very helpful for our future work.
In this revision, we cited Painter's study and some similar studies on permafrost process modeling, in which CHT is incorporated into the energy balance, in the Introduction section:
*"A number of traditional schemes for soil heat transport have been further developed with enhanced vapor/liquid CHT processes and have been shown to be effective in cold regions (He et al., 2018; Kurylyk et al., 2014; Wang and Yang, 2018).Furthermore, researchers have recently begun to formulate soil heat and water transport processes within a three-dimensional framework to provide a more reasonable physical expression for vertical and horizontal heat and mass transport (Orgogozo et al., 2019; Painter et al., 2016)."*
Further, we have discussed the advantages of a 3-D LSM over the 1-D one in the Discussion section 4.3. By coupling both horizontal and vertical thermal and hydrological processes at the surface/subsurface, those 3-D models make them advantageous over 1-D model like the SHAW model in representing the surface/subsurface water migrations and the thermal consequences on permafrost.

*"In addition, Kurylyk et al. (2016) mentioned the potential thermal impacts caused by lateral discharges in permafrost regions, while relatively small. Unfortunately, it is not investigated in this study because the one-dimensional SHAW model ignores lateral water migration from the perimeter into the soil column due to soil anisotropy, which may lead to some uncertainty in simulating water flow within the active layer. Recently, 3-D LSMs have emerged as found in the recent literature (Endrizzi et al., 2014; Painter et al., 2016; Rogger et al., 2017). Those models couple both horizontal and vertical thermal and hydrological processes at the surface/subsurface and own obvious advantages over 1-D models in terms of the physical basis for studying surface/subsurface water migrations and the thermal consequences on permafrost. Although current observations on the QTP cannot meet high data and parameter demands of 3-D models for, this provides another direction for CHT studies on the QTP. "*

**RC1:** Is $q_l$ a Darcy flow or water head potential in the current formulation? My understanding was that the current model considers both saturated and unsaturated soil conditions. It would be nice to include the discussion on the compute time difference between saturated and unsaturated cases.

**Authors:** $q_l$ is determined by the hydraulic conductivity and water head potential difference between two soil layers. We have provided Eqs. (3-6) about liquid water migration in Section 2.1 of the revised manuscript to avoid possible confusion. If I understand correctly, the SHAW model does not separately treat saturated and unsaturated soil conditions. Rather, it considers the saturated flow as a special case of unsaturated condition with saturated hydraulic conductivity and water potential.

$$"K = K_s \left(\frac{\Psi_e}{\Psi}\right)^{\left(2+\frac{3}{b}\right)} \tag{3}$$

*where $K_s$ is the saturated hydraulic conductivity (cm ·h-1), $b$ is an empirical parameter representing pore size distribution, and $\Psi_e$ is the air entry potential (m) for the saturated soil layer. $\psi$ is computed as a function of soil moisture:*

$$\Psi = \Psi_e \left(\frac{\theta_l}{\theta_s}\right)^{-b} \tag{4}$$

*where $\theta_l$ and $\theta_s$ are the liquid water content (m3 ·m-3) and soil porosity (m3 ·m-3), respectively. Then, the vertical water flux $q_l$ could be calculated by the water potential difference and the relative conductivity $K_{n,n+1}$ between layer n and n+1:*

$$K_{n,n+1} = (K_n \cdot K_{n+1})^{\frac{1}{2}} \tag{5}$$

$$q_l = \frac{K_{n,n+1}}{z_{n+1}-z_n}(\Psi_n - \Psi_{n+1} + z_{n+1} - z_n) \tag{6}"$$

**RC1:** What is the equation for the vertical water flux? It is not clear how one would define subsurface water transport without addressing the soil permeability?

**Authors:** The equations related to soil ice content and soil permeability due to ice have been added as Eqs.(7-11) in the revised manuscript:

*"If the soil temperature is below 0 ° C and ice is present, the total soil water potential is estimated using a modified Clausius-Clapeyron equation:*

$$\emptyset = \Psi + \pi = \frac{L_f}{g}\left(\frac{T}{T_K}\right) \tag{7}$$

*where $\emptyset$ is the total soil water potential, $\Psi$ is the matric potential from Eq. 4, $\pi$ is the osmotic potential (m) with respect to solutes in the soil, $g$ is the acceleration due to gravity (m·s-2), $L_f$ is the latent heat consumption during phase change (kJ·kg-1), and $T_K$ is the freezing point (°C). Thus, the unfrozen water content and ice content can be solved by combining Eq. 4 and Eq. 7:*

$$\theta_l = \theta_s \left(\left(\frac{L_f T}{g T_k} - \pi\right)/\Psi_e\right)^{-\frac{1}{b}} \tag{8}$$

$$\theta_i = (\theta_w - \theta_l)\frac{\rho_i}{\rho_l} \tag{9}$$

*where $\theta_w$ is the total water equivalent in the soil layer. When ice content is present, hydraulic conductivity is inhibited:*

$$K_i = \begin{cases} 0, & p - \theta_i < 0.13 \\ f \cdot K, & p - \theta_i \geq 0.13 \end{cases} \tag{10}$$

*where $p$ is the available porosity, and $f$ is a fraction for linearly reducing the soil hydraulic conductivity:*

$$f = \frac{p - \theta_i - 0.13}{p - 0.13} \tag{11}"$$

According to these equations, soil permeability in SHAW model is jointly limited by ice content, soil texture and hydraulic characteristics such as porosity and hydraulic conductivity. When the soil freezes and the available porosity is less than 0.13, the soil permeability is set to 0, and when the available porosity is more than 0.13, vertical flux is allowed but limited.

**RC1:** I suggest including the permeability paragraph in the Discussion section. During snowmelt, the surface is typically frozen. That said, most if not all of the snowmelt water should runoff the surface. Not clear why it would percolate into the subsurface? Westermann et al., has a publication on the effect of water percolation in the snow layer and its impact on surface temperatures, worse to mention as well.

**Authors:** Thanks for this very important comment. According to our experiments, there can be two effects of snowmelt relating to convection on soil temperature. One is related to infiltration of snowmelt water into the soil, and the other is related to an indirect effect due to the change in the surface temperature gradient, which then alters the temperatures of the underlying soil. The latter effect can occur even when the surface is frozen and no liquid infiltrates into the soil.

The first effect is apparently observed during the spring melting periods, when the snow is melting and the soil may be not yet be completely thawed. As shown in Eq.10 and 11 in the revised manuscript, the SHAW model allows liquid water migrates between soil layer when ice content is low. Therefore, in spring, even if the soil layer is not yet completely thawed, spring snowmelt can slowly infiltrate and carry convective heat to the soil layers.

The second effect is the one you are concerned with. We observed such an effect in the January 2010, when the soil layer was completely frozen and impermeable, but some small snowmelt events affected the temperature of the shallow soil layers. On those occasions, we found that convective heat transfer (CHT) does not directly affect the soil thermal regime by infiltrating the liquid flow, but indirectly affect that soil layers by altering the thermal gradient at the surface. In this case, although the snowmelt stopped at the ground surface, it altered the temperature at 0.0 m depth and affected the thermal gradient at shallow depths. Then these temperature deviations spread to the lower depths through the changed thermal gradient. Although this type of heat transfer is accomplished through heat conduction, it is still essentially an indirect convection-induced heat transport. In our experiments, we disabled the CHT and retained water flow and heat conduction in the modified models. Therefore, warmer snowmelt from the snow surface moves downward to the ground surface by the original model than by the modified models. Our experiments can precisely produce this indirect effect associated with snowmelt.

Accordingly, we have added a new section, *"Section 4.2 Snowmelt influence on convective heat transfer"*, to discuss the snowmelt-related CHT issues. We would like to show the section here for your reference:

*"Snowmelt is a main component of spring infiltration that causes CHT from the ground surface and warms the underlying soil layers, as evidenced by both our study and the observation-based studies mentioned above. Normally, as air temperature warms, snowmelt is accompanied by thawing of the soil, allowing meltwater to freely infiltrate into the soil. This is the main form of snowmelt-induced CHT. However, as shown in our contrasting experiments, the soil temperature of the Control scenario differs from that of the NoSurf and NoConv scenarios even during the periods when no vertical water flux occurs in these layers. This suggests that snowmelt could also have an indirect influence on the soil without explicit infiltration. For example, some snowmelt events took place in January 2010 (Figure 3), possibly due to a transient increase in air temperature or shortwave radiation, resulting in temperature differences between Control and NoSurf/NoConv at shallow depths (Figure 4). However, snowmelt could not infiltrate because the ground was still in an impermeable, frozen state at this time. We assume that in this case the temperature at the ground surface was affected by the convective heat carried with snowmelt, although snowmelt can only move downward through the snowpack and reach the ground surface before it drains laterally, which then affects the thermal gradient at shallow depths. Thus, the altered temperature at the ground surface was spread to the lower layers by the changed thermal gradient. Not coincidentally, we also measured some CHT-induced soil temperature changes at intermediate depths, but at the same time no corresponding convective fluxes were observed, which we believe is also part of the indirect convective heat influence exerted from other depths to this depth by conduction. Although this type of heat transfer is accomplished through heat conduction, it is still essentially an indirect convection-induced heat transport."*

We also cited two related studies by Westermann (Magnin et al., 2017; Zweigel et al., 2021) to discuss water redistribution within snow and its impacts on ground temperature at Section 4.2:

*"Moreover, percolation of liquid water within snow leads to a complex spatial redistribution of snow depths and densities, which strongly regulates the ground temperature and the active layer thickness in snowpack areas (Magnin et al., 2017). Zweigel et al. (2021) have reported that redistribution of snow, taking into accounts snow water percolation, increased ground surface temperature 1-2 °C, demonstrating another aspect of the indirect impacts of snow water migration on the permafrost thermal regime."*

**RC1:** I found Figure 3 not necessarily helpful for understanding the effect of convective heat flow. Does it mean that convective flux adds to the 5C difference in soil temperatures over a short period of time?

**Authors:** I apologize for not well explaining the importance of Figure 3 in the initial manuscript (now Figure 4 in the revised manuscript because we added new Figure 1 showing the time series of air temperature and precipitation during the study period). Figure 4 is an overall demonstration of the impacts of CHT, presenting not only the soil temperature deviations due to CHT, but also the characteristics of its occurrences in time and at depths. As shown in Figure 4, it is clear that CHT usually works during the spring when snow cover and soil ice begin thawing, and the deeper soil layers are later affected by CHT than the shallow layers. Moreover, Figure 4 also shows that the intensity and frequency of CHT in shallow layers are significantly higher than in intermediate and deep layers. The large difference between shallow, intermediate and deep depths indicates the need for further analysis at different depths respectively. By considering the above reasons, we believe Figure 4 is important to understand the role of CHT in affecting permafrost thermal regime. We have strengthened the importance of Figure 4 at Section 3.2:

*"The differences in the soil temperature profiles between the Control and the two other scenarios, i.e., partial (NoSurf) or full (NoConv) exclusion of CHT in the model, are presented in Figure 4b and Figure 4c, respectively, which depict the distribution patterns of CHT occurrence in time and depth, and the intensity of soil temperature variations due to CHT."*

We also improved the legend of Figure 4b and c in order to more clearly express the temperature deviation ranges.

**RC1:** Can the changes that are shown in Figs 3 and 4 be verified against observations or in lab measurements? L353 concludes that convective flux could contribute to up to 10C temperature change during snowmelt. This is massive warming or cooling, which I am having a hard to believe. Short-term 10C warming/cooling should lead to substantial

changes in subsurface thermal state over time. Are there any pieces of evidence of the accelerated ground warming in the region?

**Authors:** Many thanks for your suggestions. Some existing studies have reported such soil warming events due to snowmelt or precipitation. We have added them as cross-validation evidence in the Discussion section 4.1:

*"Previous studies have also observed some CHT effects due to the anomalous fluctuations in soil temperature during the periods when snowmelt or rainfall infiltrates into the soil. Kane et al. (2001) and Hinkel et al. (1996) reported the step-like increase in near surface temperature by 2 °C to 4°C and the pronounced disruption of thermal gradients during the periods when snowmelt infiltrates. The increase in soil temperature (Iijima et al., 2010; Mekonnen et al., 2021) and frost front depth (Douglas et al., 2020; Guan et al., 2010) was also observed following heavy rainfall events, indicating that precipitation is another important source of CHT in addition to snowmelt. Kane et al. (2001) estimated that the CHT during heavy precipitation is twice as high as conductive heat. Hinkel et al (2001) measured 0.5 °C and 1.3 °C positive changes in soil temperature in response to infiltration from snowmelt and rainfall, respectively."*

According to these observation based studies, CHT due to snowmelt and precipitation usually alters the soil temperature at a range of 0 to 4 °C in several hours, and promotes the thawing of active layer. Although the maximum temperature deviations due to the CHT could reach 10 °C in our study, most of the deviations were limited to less than 5 °C, which is consistent with the observations. In addition, as already stated in the Introduction, although these observed soil temperature warming are mainly driven by CHT, they are still produced by joint effects including heat conduction, advection and convection, and phase change. It is still hard to isolate the sole impacts of CHT from the totality of soil heat regime by observations. Thus, our modelling study provides a good explanation for the mechanism behind these observations, how those warming events at certain depths during spring thaws and rainfall events could occur due to CHT processes alone.

We have strengthened the comparison between previous observations and our results by adding some related discussion in the Discussion section 4.1:

*"Although the maximum temperature perturbation due to the CHT process could occasionally reach 10 °C in our study, most of the temperature perturbations were limited to less than 5 °C, which is consistent with the observed phenomenon. Although these observed warming events are mainly driven by liquid convection, they are still a combined product of multiple heat transfer processes, including conduction and vapor flow. On the other hand, our modelling study provides a good explanation for the mechanism behind these observations, how those warming events at certain depths during spring thaws and rainfall events could occur due to CHT processes alone."*

**RC1:** I suggest quantitively estimate the impact of these short-term warming/cooling pulses on mean annual ground temperatures. Are these pulses significant, and on what temporal scale? For example, on the yearly scale, those pulses could be less important than on a monthly scale and so on....

**Authors:** Thank you very much for this important comment. As per your suggestion, we singled out each CHT events, which is defined as a soil temperature deviation of more than 0.1 °C between Control and NoSurf/NoConv at a time step, and quantify their effects. A temperature deviation of 0.1 °C or less is insignificant and might be caused by model iteration bias rather than CHT.

Despite CHT are relatively slight on the annual scale, we found it could significantly affect soil temperature at the time in which CHT is happened. The total amounts of heating and cooling CHT events and the mean temperature deviations per each CHT event were analyzed for measuring the frequency and magnitude of CHT effects on soil temperature. We revised the method description as presented in Section 2.2:

*"We defined a CHT event as a ground temperature deviation of more than 0.1 °C between NoSurf/NoConv and Control at one model time step. A deviation of 0.1 °C or less is trivial and could be due to model iteration bias rather than CHT. According to their effect, there are cooling CHT events and heating CHT events. The total numbers of cooling and heating CHT events and mean temperature deviations were analysed to examine the frequency and magnitude of CHT effects on ground temperature:*

$$\overline{\Delta T} = \frac{\sum_{i=1}^{m} \Delta T}{m} \; (12)$$

*where, $\overline{\Delta T}$ is the mean temperature deviation of all heating or cooling CHT events, $\Delta T$ is the temperature deviation caused by an CHT event, and m indicates the count of heating or cooling CHT events."*

Our results were presented in a newly added Table 2:

**Table 2   The numbers of heating and cooling CHT events and the mean temperature deviations caused by CHT of 0.05 m, 1.05 m and 2.45 m depths. The deviations of 0.1 °C or less were excluded for statistics.**

|  | Control vs. NoSurf | | | Control vs. NoConv | | |
|---|---|---|---|---|---|---|
|  | 0.05 m | 1.05 m | 2.45 m | 0.05 m | 1.05 m | 2.45 m |
| Number of heating events | 2436 | 1850 | 602 | 3109 | 2984 | 456 |
| Average increase (°C) | 0.86 | 0.43 | 0.211 | 0.972 | 0.412 | 0.229 |
| Number of cooling events | 1195 | 189 | 10 | 1757 | 1302 | 67 |
| Average decrease (°C) | -0.79 | -0.24 | -0.200 | -1.06 | -0.407 | -0.200 |

Table 2 shows that the heating effects of the CHT from surface infiltration (Control vs. NoSurf) are nearly the same as the CHT from both surface infiltration and migration between soils (Control vs. NoConv) in terms of frequency and mean increased temperature at each depth, indicating that infiltrative liquid flux from ground surface is the main source of the heating CHT and usually causes a mean temperature increase of about 0.9 °C, 0.4 °C, respectively, in 0.05 m and 1.05 m depth. While the Control vs. NoConv shows more frequent and drastic temperature decrease than Control vs. NoSurf, especially in 1.05 m depth (1302 cooling events and –0.407 mean temperature deviation in Control vs. NoConv, but only 189 cooling events and -0.244 mean temperature deviation in Control vs. NoSurf), which highlights that water flux between soil layers is more frequent in cooling the soil temperature.

Accordingly, we revised Section 3.3.1 about the shallow depths:

*"The convective heat estimated in Control acts as an extra heat source during the spring melting periods when CHT occurred, the source that not only provided heat for the phase changes from ice to water, but also warmed the soils and caused an average increase in soil temperature of about 0.9 °C (shown in Table 2), compared to the NoSurf and NoConv results. The maximum temperature warming could reach 10 °C at a certain time."*

*"Apart from the imposed heating effect, an opposite cooling effect is observed at shallow depths, indicated as negative differences in Figure 6a and Figure 7a. Soil temperature decreased by an average of -0.79 °C and -1.06 °C (Table 2), respectively, when contrasting the results of Control with those of NoSurf and NoConv during the spring thaw and fall freeze periods, with extreme temperature reduction by up to -5 °C occurring in some durations."*

*"Figure 4 already shows that more cooling events were triggered by convective processes within the soils than by infiltration. It becomes even clearer by comparing Figure 7a showing Control-NoConv with Figure 6a showing Control-NoSurf. Table 2 shows that there are 1757 cooling events in the comparison between Control and NoConv, but only 1195 between Control and NoSurf."*

We also revised Section 3.3.2 about the intermediate depths:

*"Temperatures at these depths averaged 0.43 °C (Table 2) higher in Control than in NoSurf during thaw periods when CHT events occurred."*

*"In these cases, temperatures in NoConv surpassed those in Control by an average of 0.41 °C for each cooling event (Table 2), while temperatures in NoSurf surpassed those in Control by only an average of 0.24 °C (Table 2). The frequency of cooling CHT effects when comparing Control and NoConv (1302 times over the entire simulation period) was also several times higher than when comparing Control and NoSurf (189 times), indicating that liquid flux between soil layers exerts more cooling effects on soil temperature at intermediate depths than infiltrative flux does."*

At last, we also improved the Abstract and Conclusions by replacing the maximum temperature disruption with the mean values.

**RC1:** Taking into consideration the above comments and questions, the paper needs more clarification on the mathematical formulation, application, and discussion with similar studies.
Authors: Thanks.

**RC1: Minor**
L45-48 …influenced. Influenced by what? Not clear. I suggest to re-write.
**Authors:** Done, this sentence has been re-written as:

*"Given the current warming trends in most of the Earth's permafrost areas (Biskaborn et al., 2019), significant changes in permafrost dynamics are likely to occur, and the local ecosystem and environment have already been seriously influenced by regional hydrological and thermal changes caused by permafrost degradation (Cheng and Wu, 2007; Jin et al., 2009; Jorgenson et al., 2001; Tesi et al., 2016)."*

**RC1:** L121-124 need a reference.

**Authors:** Done. We also have provided some more details about the reasons why the SHAW model outperforms the other LSMs, one is the detailed physics of CHT process and the other is the special iteration scheme for reducing the biases due to its complex physic basis:

*"The SHAW model is one of the few land surface models (LSMs) that considers the detailed physics of the interrelated mass and energy transfer mechanisms, including precise convective heat transport processes of liquid water and vapor (Kurylyk and Watanabe, 2013), making it advantageous for demonstrating the important interactions between soil water dynamics and frozen soil thermal regimes in permafrost regions (Flerchinger et al., 2012). In addition, SHAW applies a special iteration scheme in which a time step is subdivided into multiple sub-time steps to control the error from the previous step in solving the mass and energy balance and to strictly enforce the mutual coupling of the hydrological and thermal processes (Flerchinger, 2000)."*

**RC1:** In Table 1. Are any of the parameters been tuned during this study?

**Authors:** In this study, we used a Latin Hypercube sampling method to generate 1000 combinations of parameters, then we examined parameter sensitivity and identified the optimal values for the sensitive parameters for the Control (the original SHAW model) experiment. Only 4 parameters (i.e., saturated hydraulic conductivity, air-entry potential, saturated volumetric moisture content, and pore-size distribution index) were calibrated and the others were assumed to be the default values. The calibrated parameters in Control were consistently applied to the other two scenarios to make sure the parameter value won't influence the results. Since the Control scenario takes the full consideration of CHT and the other two are incomplete, comparing their outputs based on the same parameter values can filter out the sole CHT impact from the interrelated thermal transfer system. We have improved the description about this issue in Section 2.4:

*"The most optimal parameter values from the 1000 combinations, as presented in Table 1, were consistently applied to all three scenarios designed in Section 2.2 to eliminate the influence of parameter values on the inter-scenario comparison."*

In this revision, we also clarified how to do with the model parameters and their uncertainties, as requested by the other reviewer:

[revised manuscript text omitted]

**RC1:** On Figure 2. It is hard to tell the time of snowmelt.

**Authors:** We have replaced the Figure 2 in the initial manuscript (now Figure 3 in the revised manuscript) with a new figure in which the specific dates of snowmelt events have be mentioned. The new figure is presented here for your reference.

[Figure]

Figure 3    Unusual warm air temperature in January 2010, presumably related to overestimation of modelled soil temperature that month, and simulated hourly snowmelt, also influenced by air temperature. The dates (month/day) at the top of the figure indicate when the snowmelt events occurred.

**RC1:** Additional language improvements are needed as well.

**Authors:** Done. We revised the manuscript very carefully and we think the language has been much improved.

---

## Author Comment (AC2)

**Response to Reviewer #2:**

We would like to thank Reviewer #2 for these valuable comments. Below are the responses to all Reviewer #2's comments and how we address them in the revised manuscript. **Texts in red** are the reviewer's comments; **those in black** are the authors' explanations to the reviewer's comments; and *those in blue* are the revised texts in the revised manuscript.

**RC2:** This manuscript applied the SHAW model to investigate the impacts of CHT on active layer thermal dynamics on the Tanggula station, a typical continuous permafrost site located at the eastern Qinghai-Tibetan Plateau with abundant meteorological and soil temperature/moisture observation data. The 2008-2009 observed hourly data were used to calibrate the model parameters and those of 2010 for validation. The control experiment was carried out to quantify the changes in active layer thermal regime affected by vertical advection of liquid water, consisting of three setups: using (1) the original SHAW model with full consideration of CHT; (2) a modified SHAW model ignoring the CHT due to infiltration from the surface, and (3) a modified SHAW model ignoring complete CHT processes in the system. The results show that the CHT events mainly happened during thawing periods when the active layer melted at shallow (0-0.2m) and middle (0.4-1.3 m) soil depths, and its impact on soil thermal regime at shallow depths was significantly greater in spring melting periods than in summer. The impact was minimal in freezing periods and in deep soil layers. The overall annual effect of CHT by liquid flux is to increase soil temperature in the active layer and accelerate the thawing of permafrost at the study site. The topic is interesting and suitable for the journal The Cryosphere. I do have some major concerns such as: The introduction should be consistent with the topic. The discussion should be stated as different format with the results. The concepts such as the difference between unfrozen water and liquid water content need to be defined in the context.

**Authors:** Thank you for your kind comment. We have improved the Introduction and restructured the discussion section. Please refer to the responses below for details. Unfrozen water and liquid water have been clearly distinguished in this revised manuscript. Unfrozen water refers to the residual liquid content remaining in frozen soil, while liquid water refers to the water content during thawed periods when there is no ice in the ground. However, throughout the figures we consistently refer to it as unfrozen water content (UWC), which is also common in many research papers on frozen soils.

**RC2:** I listed my major concerns as follows:
1.The abstract is too long and needs to be shorten.
**Authors:** Thanks, we tried to shorten the abstract without missing the key findings.

**RC2:** 2.My main concern is that the data in 2008-2010 was used in this study, which was too old. I recommended to update the time series of the data.
**Authors:** We have collected the time series of the TGL data from 2007 to 2017. Unfortunately, among them only the climate data from April 2007 to December 2010 are on the hourly time scale, and the others are on the daily scale. Since CHT is a rapid heat exchange process that usually occurs within a few hours, and the SHAW model is not

satisfactory to run at a daily time scale, we had to drop the daily data to avoid potential errors in the simulation.

More importantly, in this study we are interested in the impacts of CHT on the permafrost thermal regime, rather than the temporal variation of CHT. We believe that the choice of the experimental periods does not significantly affect our results as long as the data are reasonable and highly quality-controlled.

The three-year experimental period in use includes two years with significant CHT impacts (2008 and 2010) and one year with relatively weak impacts (2009), which could support the investigation of interannual differences of CHT and the influencing factors associated. For the above reasons, we are unable to update the time series of climate data.

However, in response to this concern, we justified our choice of the experimental data and strengthened some discussion about the lack of high-quality data in Section 4.4:

*"CHT is a rapid thermal exchange process that generally occurs at hourly intervals and is influenced by soil moisture, so hourly data of high quality are used for CHT study. However, due to the harsh natural environment and cumbersome transport, we are unable to obtain a large amount of long-term, high-quality climate and permafrost observation data from multiple sites. Though we have conducted a three-year long-term experiment at the TGL site that included two years of significant CHT impacts (2008 and 2010) and one year of relatively weak impacts, which could support investigation of interannual differences of CHT and the influencing factors associated, we still face the lack of high-quality data to enable a spatial and longer investigation into the CHT impacts. The good news is that some new observation sites have been deployed during the ongoing campaign of the second Tibet expedition (Chen et al., 2021), and the data situation for CHT studies is improved."*

**RC2:** 3.“Tibetan” or “Tibet”? It should be maintained consistent throughout the text.
**Authors:** Done. We have unified them with “Tibet” in the context of Qinghai-Tibet Plateau because the former Qinghai is a place name.

**RC2:** 4.The logic of the introduction is a little bit unclear, and some of its details are not adequately rigorous. As I know, there were many works that has been done on this work based on different models, such as Yu et al., 2018, Liquid-Vapor-Air Flow in the Frozen Soil, JGR, and He et al., 2018, A coupled model for liquid water-vapor-heat migration in freezing soils, Cold Regions Science and Technology. I suggest that the authors to provide a thorough review on this aspect and state the previous pros and cons on the previous work, then clearly state that why this need be done to make a real progress and what is the different between them?
**Authors:** We have improved the Introduction from the following aspects:
(1) Some evidence of observed soil temperature warming events due to snowmelt and precipitation convection and the limitations of observation-based CHT studies have been emphasized in the revised manuscript. We argue that for in situ observation study, it is difficult to isolate the sole impacts of CHT from the totality of heat transfer processes in the soil. It justifies the use of numerical approach in this study.

*"Migration of liquid water can usually be forced by gravitational, pressure or osmotic pressure gradients in soils during thaw periods. During spring snowmelt and summer rainfall, a rapid temperature increase of about 2°C to 4°C is observed in the uppermost soil layer, indicating a heating effect of liquid CHT (Hinkel et al., 1996; Hinkel et al., 1997; Kane et al., 1991). As a result, the warming of soil temperature by liquid CHT increases the depth of thaw in frozen ground (Douglas et al., 2020; Guan et al., 2010)."*

*"Although some convective heat effects have been observed, they are often produced by simultaneous processes such as heat conduction, advection and convection, and phase changes. In-situ instrumentation is still limited to accurately measure key thermal and hydrological soil variables. It is challenging to isolate the sole impacts of CHT from the totality of heat transfer processes in the soil (Hasler et al., 2008; Pogliotti et al., 2008)."*

(2) In the initial version, we focused on the studies that directly investigated CHT, but ignored the modelling studies that considered CHT in simulation. As per your suggestion, we reviewed much of the related literature and tried to incorporate them in the Introduction section.

Recently, a few modelling studies in permafrost regions have started to incorporate vapor and liquid CHT into energy equilibrium, and the impacts of vapor and air flow on permafrost thermal regime have been provisionally explained. In this revised manuscript, the advances in LSMs development with CHT processes considered and the studies of the role of vapor CHT have been included:

*"The demand for complete modelling of permafrost changes has therefore recently prompted interest in the development of simulation tools for coupled heat transport and variable hydrological processes to account specifically for CHT. A number of traditional schemes for soil heat transport have been further developed with enhanced vapor/liquid CHT processes and have been shown to be effective in cold regions (He et al., 2018; Kurylyk et al., 2014; Wang and Yang, 2018). Furthermore, researchers have recently begun to formulate soil heat and water transport processes within a three-dimensional framework to provide a more reasonable physical expression for vertical and horizontal heat and mass transport (Orgogozo et al., 2019; Painter et al., 2016). By using these advanced models, the role of CHT on the permafrost thermal regime, especially the vapor CHT, was provisionally explained. Wicky et al.(2017) developed a numerical model considering air flow in permafrost talus slopes and revealed pronounced seasonality of the air flow cycle on talus slopes and considerable seasonal differences in the effects on soil temperature. Yu et al. (2018) and Yu et al. (2020) quantified the thermal response to different types of vapor migration associated with evaporation and air flow, respectively. Luethi et al. (2017) estimated the heat transfer efficiency of vapor and liquid convection. Kurylyk et al. (2016) developed a three-dimensional coupled soil heat and water model to investigate the effects of runoff on soil temperature."*

(3) We also highlighted the reasons why we chose the SHAW model rather than others to study the effects of CHT in the Introduction part. The details of this revision are provided in our response to .

(4) We revised the final paragraph of Introduction to further clarify the purpose of this study.

*"Therefore, in this study, the SHAW model is used to quantify the impacts of liquid CHT on the active layer temperature and moisture content through numerical modelling at a typical permafrost site, i.e., the Tanggula (TGL) site on the QTP, China. The SHAW model was modified to remove the CHT processes, and then control experiments were set up to simulate comparative scenarios with or without CHT included in the model. The specific objectives are: (1) reveal the characteristics of the CHT events in time and depth; (2) quantify the impacts of liquid CHT on the thermal regime of the active layer; (3) elucidate the interplay of heat and soil moisture during the freezing-thawing process in the active layer."*

**RC2:** 5. How representative is the Tanggula site for such a large area of the Qinghai-Tibet Plateau? It is suggested to add more points in different regional to illustrate the problem. Why did the authors select this site?

**Authors:** Thanks for this question. We selected Tanggula (TGL) site due to the following reasons:

Firstly, the TGL site is one of the few permafrost monitoring sites on the Qinghai-Tibet Plateau with long-term continuous observations of the active layer in parallel with high-quality hourly meteorological data. Among the previous studies, the data at the TGL site have been widely used in permafrost research. We have emphasized the representativeness of the TGL site in Section 2.3:

*"A typical permafrost site, the TGL site on the QTP, was selected for this study because of long-term, quality-assured observations of the active layer and deep permafrost in parallel with meteorological observations at high temporal resolution. Due to the ideal representativeness to elevation-controlled permafrost on the QTP, this site has been widely used for alpine permafrost research such as permafrost hydrothermal characteristics (Li et al., 2019), permafrost response to climate change (Zhu et al., 2017; Zhu et al., 2021), and permafrost process modelling (Hu et al., 2015; Li et al., 2020)."*

Secondly, as we responded to Question 1, the CHT process is a rapid heat transfer process that usually happens on the minutely to hourly time scale. High time resolution meteorological data and soil temperature/moisture data are necessary for study the CHT impacts on the thermal regime of the active layer in permafrost regions. In addition, the SHAW model also requires high-quality driven data and low boundary conditions at hourly time step to ensure a good simulation. Although we have collected data from other permafrost sites which are mainly distributed around the Qinghai-Tibet Highway, their temporal resolutions and data quality do not meet the requirements of this study.

Here we provided an additional experiment at another site, the Wudaoliang site near the Qinghai-Tibet highway/railway (coded QT08, located at latitude of 33°35' N and longitude of 92°52' E). The lower-boundary depth of QT08 was at 2.4 m with daily observed soil temperature and moisture. The soil column stratification and texture were extracted from

one of our previous work, Wu et al. (2018). Since no hourly meteorological data are available for this site – only daily are available, we extracted the meteorological forcing from the ITPCAS CMFD gridded meteorological forcing dataset (0.1° spatial resolution and **3 h time** resolution). This CMFD is believed to have good accuracy in particular for the eastern QTP and is widely used for regional permafrost modeling on the QTP.

Following the same modeling settings, we simulated the Control scenario (full presence of CHT) and NoConv scenario (complete absence of CHT) for our contrasting experiment. We counted the number of heating and cooling CHT events (a CHT event is defined as a ground temperature deviation between Control and NoConv at one model time step) and analyzed mean temperature deviations in terms of the frequency and magnitude of CHT effects.

Figure S1 (below) presents the simulated soil temperature of QT08 under Control and the temperature difference between Control and NoConv. Table S1 shows the number of heating and cooling CHT events and the mean temperature deviations at QT08. Served as a comparison, we also provide the results at the TGL site in Table 2, which is also newly added to the revised manuscript as per the comment of another reviewer.

We found that at QT08, the major role of CHT is still to heat the soil during spring thawing, along with a small number of cooling effect, which is consistent with the findings at TGL site. The patterns of CHT effects in temporal and at depth at QT08 are also similar to those at TGL (Figure S1 and Figure 4 in the revised manuscript). However, the temperature deviations caused by CHT at QT08 are much smaller (-4 to 4 °C with an average of about 0.1 °C) than those at TGL (-5 to 10 °C with an average of about 0.9 °C), especially in shallow layers. We believe that the 3 h modeling time step diluted the thermal influence of hourly rapid soil water migrations at shallow depth. In addition, due to the mismatching between 0.1° gridded forcing data and in-situ active layer observations, the model performance in QT08 (Figure S2) was not so good as in TGL site. The simulated soil temperatures were overall overestimated compared to observed values. Due to a large number of abnormal 0 values in observed UWC time series at QT08, we do not show the comparison between observed and simulated UWC here. After careful consideration about the relatively poorer results of QT08, we would like to show this additional experiment here only for your information rather than include it to the revised manuscript.

Considering the requirements for high-precision meteorological and soil data for this study, we have almost no options other than the TGL site. Recently, some new permafrost observation sites have been deployed on the Qinghai-Tibet Plateau with an intention of long-term monitoring. Although we cannot access those site data right now, we believe that soon we will be able to use them to extend the CHT studies. We have also supplied some discussion about the data scarcity problem as can be found in the response to Question 1.

[Figure]

**Figure S1** Simulated soil temperature profiles of QT08 under the Control scenario (a), and the differences in soil temperature between the scenarios Control-NoConv (b).

[Figure]

**Figure S2** Simulated and observed daily soil temperatures at the QT08 site during 1 January 2015 to 31 December 2018

**Table S1** The numbers of heating and cooling CHT events and the mean temperature deviations caused by CHT at QT08 permafrost observation site.

| | Control vs. NoConv (QT08) | | |
|---|---|---|---|
| | 0.07 m | 0.66 m | 1.84 m |
| heating events number | 1582 | 2094 | 770 |
| mean increased temperature (°C) | 0.114 | 0.162 | 0.166 |
| cooling events number | 591 | 980 | 1502 |
| mean decreased temperature (°C) | -0.143 | -0.121 | -0.123 |

**Table 2** The numbers of heating and cooling CHT events and the mean temperature deviations caused by CHT of 0.05 m, 1.05 m and 2.45 m depths. The deviations of 0.1 °C or less were excluded for statistics.

| | Control vs. NoSurf | | | Control vs. NoConv | | |
|---|---|---|---|---|---|---|
| | 0.05 m | 1.05 m | 2.45 m | 0.05 m | 1.05 m | 2.45 m |
| heating events number | 2436 | 1850 | 602 | 3109 | 2984 | 456 |
| mean increased temperature (°C) | 0.861 | 0.430 | 0.211 | 0.972 | 0.412 | 0.229 |
| cooling events number | 1195 | 189 | 10 | 1757 | 1302 | 67 |
| mean decreased temperature (°C) | -0.785 | -0.244 | -0.200 | -1.06 | -0.407 | -0.200 |

**Reference:**

**Wu, X., Nan, Z., Zhao, S., Zhao, L., Cheng, G.: Spatial Modeling of Permafrost Distribution and Properties On the Qinghai-Tibet Plateau, Permafrost and Periglacial Processes, 29, 86-99, https://doi.org/10.1002/ppp.1971, 2018.**

**RC2:** 6.Why to use SHAW model, the introduction is not clear.

**Authors:** We chose the SHAW model for this study due to its two advantages:

(1) On one hand, the SHAW model is one of the few land surface models (LSMs) in which the thermal and hydrological processes is well coupled. SHAW considers the heat transfer by hydrological processes including liquid water flow and vapor/air flow, which makes it outperform than other LSMs in simulating and investigating the relationship between water cycle and energy exchange. In this respect, SHAW is suitable for our study on how liquid CHT affects the thermal state of permafrost active layer.

(2) On the other hand, the SHAW model applies a special iteration method to solve the energy and mass balance matrix. Generally, an LSM iterates only once in a time step, which may lead to some bias because the soil moisture or temperature values from the previous time step will directly join mass and energy balance iteration at current time step without correction. It may lead to error accumulation and propagation during the numerical computation. The SHAW model adopts a smart approach to reduce this kind of error. If the energy/mass change $\Delta$ between two adjacent time steps exceeds a given threshold (0.001 or smaller), the time step will be subdivided into many sub-time steps and re-iterating. When the $\Delta$ between each sub-time step is less than the threshold, the moisture or temperature from the previous sub-time can be considered as unchanged in this sub step and could directly be used in current iteration. Through this iterative approach, the SHAW model strictly enforces the mutual coupling of hydro and thermal processes. One drawback relating to such treatment is a demand of accurate lower boundary, which is provided by the observed soil temperatures in this study.   An accurate simulation is precondition for investigation of CHT in this study.

Since the first point was already present in the previous version, we have strengthened the second point to the Introduction section of revised manuscript:

*"The SHAW model is one of the few land surface models (LSMs) that considers the detailed physics of the interrelated mass and energy transfer mechanisms, including precise convective heat transport processes of liquid water and vapor (Kurylyk and Watanabe, 2013), making it advantageous for demonstrating the important interactions between soil water dynamics and frozen soil thermal regimes in permafrost regions (Flerchinger et al., 2012). In addition, SHAW applies a special iteration scheme in which a time step is subdivided into multiple sub-time steps to control the error from the previous step in solving*

*the mass and energy balance and to strictly enforce the mutual coupling of the hydrological and thermal processes (Flerchinger, 2000)."*

*"The fine consideration of CHT processes, the mutual coupling of hydrothermal processes, and the broad applicability render the SHAW model capable of investigating the impacts of CHT on permafrost thermal regimes."*

**RC2:** 7. In model settings: Table 1 should remove to this section, and what I want to know is how this parameter is obtained, which has important effects on the simulation results.

**Authors:** Thanks for this important comment. We have removed Table 1 to Section 2.4. In response to Question 7 and 8, we have complemented a new parameter calibration experiment to identify parameter sensitivity and model uncertainty associated.

According to prior knowledge and the variables involved for CHT, we calibrated four main hydraulic parameters, i.e., saturated hydraulic conductivity, air-entry potential, saturated volumetric moisture content, and pore-size distribution index. We used a Latin hypercube sampling method to do this. The steps are: (1) 1000 independent combinations of the four parameters were randomly generated within the priori parameter ranges by the Latin hypercube sampling method; (2) the 1000 combinations were applied to drive the model one by one and their outputs were compared and evaluated against observed data in order to determine the optimal parameter values for each layer; (3) the 95% probability bands (95PPU) of the model outputs of the all 1000 combinations that represent the range of distribution of model outputs due to parameter freedom were counted for analyzing model uncertainty and biases.

The detailed descriptions about this method have been included in revised manuscript in Section 2.4:

*"We calibrated the four main hydraulic parameters (Table 1), i.e., saturated hydraulic conductivity, air-entry potential, saturated volumetric moisture content, and pore-size distribution index, relating to soil moisture in the model, while keeping the other soil parameters as default values. Data from 2008-2009 were used for calibration and 2010 for validation. The model was run with an hourly time step and the results were then aggregated to a daily scale to facilitate comparisons and analyses. The ranges of hydraulic parameter values were roughly determined with reference to previously studies (Chen et al., 2019; Wu et al., 2018; Liu et al., 2013). To find the best parameter combination and measure model uncertainty, 1 000 independent parameter combinations randomly generated by the Latin hypercube sampling method in conjunction with the priori ranges. We restricted the values of sampling parameters in adjacent layers to assume that adjacent soil layers have similar textures. Then the 1000 combinations were used to drive the model one by one, and their outputs were compared and evaluated to determine the optimal parameter values for each soil layer. Two metrics, including the Nash-Sutcliffe efficiency coefficient (NSE) and root mean square error (RMSE), were used to quantify the performance of the parameter combinations:*

$$NSE = 1 - \frac{\sum_{t=1}^{N}(O^t - M^t)^2}{\sum_{t=1}^{N}(O^t - \bar{O})^2} \quad (13)$$

$$RMSE = \sqrt{\frac{1}{N}\sum_{t=1}^{N}(O^t - M^t)^2} \quad (14)$$

*where $O^t$ and $M^t$ are the observed value and simulated value at time step $t$; $\bar{O}$ is the mean of the observations over the entire period; and $N$ is the total number of time steps. Considering the interaction between soil temperature and soil moisture in a coupled system, the simulation accuracy of both variables is mutually suppressed, i.e., while the accuracy of one variable continues to improve by continuously optimizing its parameter value, the accuracy of the other decreases. Thus, we determined the optimal parameter combinations by balancing the performances for both soil temperature and moisture. In addition, the 95% probability bands (95PPU) of simulated soil temperature and moisture of all 1000 random parameter combinations were also counted, showing the range of distribution of results due to parameter degrees of freedom, to measure model uncertainty introduced by parameter selection at the TGL site."*

The model outputs of the 1000 parameter combinations show that all the four parameters have important effects on simulation results. Among them saturated hydraulic conductivity was highly corelated to both soil temperature and moisture simulation, while saturated volumetric moisture content mainly controlled soil moisture. Air-entry potential and pore-size distribution index also influence the model outputs but not as pronounced as the others. Since parameter sensitivity analysis is not the focus of this study, we only included some relevant results as appropriate in Section 3.1:

*"According to our experiments, saturated hydraulic conductivity is the most important parameter that effects the simulated soil temperature."*

*"Saturated hydraulic conductivity and saturated volumetric moisture content were identified as the most important parameters controlling simulated UWC and were treated carefully."*

Based on those results, we clarified how to perform the calibration for the Control experiment and the same parameter values are used in other experiments. Please refer to the modified sections: 2.4 Model settings and 3.1 Model evaluation.

**RC2:** 8.The simulation error is large, especially for soil moisture (Fig. 1), which will bring large uncertainty to the simulation results of sensitivity experiments in this paper, and the results are doubtful.

**Authors:** As partially mentioned in our reply to Question 7, we have further analyzed model uncertainty associated with parameter degrees of freedom by counting the 95PPUs of the outputs of the 1000 random parameter combinations and shown the results on the revised Figure 2 (attached below, previously Figure 1 in the initial manuscript). The results demonstrate that the SHAW model well captures the seasonal freeze-thaw variability in both soil temperature and soil moisture, but had large uncertainty in predicting the magnitude of unfrozen soil water content, especially in the intermediate layers. The errors are considered to come from both model uncertainty and observation uncertainty.

As presents in the new Figure 2, the 95PPUs of simulated soil UWC with 1000 parameter combinations were overall above the observations at 0.4 m and 1.05 m depth (Figure 2h and i), indicating that whatever the parameter values were, the SHAW model systematically

overestimated the UWC at these depths. For correcting this systematic error, the **optimal parameter combination which could achieve lower UWC as well as ensure the good accuracy of the simulated soil temperature were specifically picked for these layers**. Some descriptions about model uncertainty and how we dealt with it were added in Section 3.1:

*"Overall, both the 95PPUs and the optimal outputs confirm a good capability of the SHAW model to simulate the complex freezing and thawing processes in the active layer given reliable lower boundaries. Seasonal variations of both soil temperature and soil moisture in the active layer of the TGL were successfully captured. The 95PPUs of soil temperature associated with the 1000 parameter combinations are narrow in band and cover the observations well at each depth, indicating the good performance and low uncertainty of the SHAW model in modelling soil temperature at the TGL site. According to our experiments, saturated hydraulic conductivity is the most important parameter that effects the simulated soil temperature. Although the 95PPUs of the simulated UWC also roughly cover the observations, a wide band and overestimation at 0.4 m and 1.05 m depths relative to the observations indicate a large uncertainty in simulating UWC and call for a necessary parameter calibration. Saturated hydraulic conductivity and saturated volumetric moisture content were identified as the most important parameters controlling simulated UWC and were treated carefully. At the intermediate depths where low liquid contents were observed, optimal parameter values are picked from the random parameter combinations for these layers that both simulate lower UWC and ensure good accuracy of the simulated soil temperature."*

Site measurement uncertainty is the other major factor that negatively affects the accuracy of the simulated soil UWC. We have noticed the abnormally abrupt declines of observed UWC at the 0.05 m and 0.1 m depths in summer of 2009 (Figure 2f, g), which were caused by equipment malfunction and decrease the simulating accuracy at these depths. Also, at the depths of 0.4 m and 1.05 m, some unrealistic zero observations of UWC were presented during the winter periods (Figure 2h, i). Considering that many studies have already affirmed a small amount of liquid pore water (ca. 0.05 m3 · m-3) continues to exist even if the soil is completely frozen (Stein and Kane, 1983), we believe that the observed UWC has systematic underestimation at these depths and our simulation results seem more realistic. We have highlighted the uncertainty from observation in Section 3.1:

*"During the summer of 2009, we noted an abrupt decline in observed UWC at 0.05 m and 0.1 m depths (Figure 2f, g), which was due to equipment malfunction. At depths of 0.4 m and 1.05 m, some unrealistic zero UWC values were also observed during the winter months (Figure 2h, i). Many studies have already affirmed that a small amount of liquid pore water (ca. 0.05 m3 · m-3) continues to exist even if the soil is completely frozen (Stein and Kane, 1983). The recorded anomalous zero values are probably related to the inadequate ability of the time domain reflectometry sensors to detect immobile residual liquid water. We believe that in these periods the simulation results appear more realistic."*

Overall, although there are still some errors now, the optimal simulation results we obtained in the revised manuscript could well capture both the soil temperature and moisture

characteristics at the TGL site and support our study on the impacts of CHT on the active layer thermal regime.

Furthermore, all the three scenarios use the same forcing and parameters, and the only difference between them is inclusion or non-inclusion of the CHT process in the model physics. Since we subtracted the results of the two scenarios with the modified models from those of the control scenario with the original SHAW model, the uncertainties that are related to the model and data, common in the three scenarios, are largely eliminated by the subtraction, and the results thus become more meaningful.

[Figure]

**Figure 2** Simulated (solid lines) and observed (dashed lines) daily soil temperatures (ST; left panels) and unfrozen water contents (UWC; right panels) at 0.05 m (a and f), 0.1 m (b and g), 0.4 m (c and h), 1.05 m (d and i) and 2.45m (e and j) depths at the Tanggula (TGL) site from 1 January 2008 to 31 December 2010. The simulated soil temperatures (solid blue line) and UWCs (solid red line) are the results with the optimal parameter values identified from the 1000 random parameter combinations. NSE: the Nash-Sutcliffe efficiency coefficient; RMSE: root mean square error. The 95PPUs of the model outputs are from all 1000 randomly generated parameter combinations.

**RC2:** 9.Why is the analysis divided into shallow, middle and deep layers, and what is the basis for the layers?

**Authors:** As shown in revised Figure 4 (attached below, previously Figure 3 in the initial manuscript), the large differences in the spatio-temporal characteristics of CHT impacts occurred between the shallow, intermediate and deep depths, which are highly related with the liquid water migration patterns at these depths. We believe that the investigation of the characteristics of CHT at different soil depths is benefitable for understanding of how convective heat transfers occurs within the soil column, and exploring the driven factors that control CHT. Thus, we selected the 0.05 m depth which is the nearest observed layer to surface, 1.05 m depth which is at the middle of the active layer, and 2.45 m depth near the low boundary, to represent shallow, intermediate and deep soil depths, respectively, and analyzed the CHT characteristics at these depths. We have strengthened the descriptions about the difference of CHT impacts between depths in Section 3.2:

*"The differences in the soil temperature profiles between the Control and the two other scenarios, i.e., partial (NoSurf) or full (NoConv) exclusion of CHT in the model, are presented in Figure 4b and Figure 4c, respectively, which depict the distribution patterns of CHT occurrence in time and depth, and the intensity of soil temperature variations due to CHT."*

[Figure]

**Figure 4    Simulated hourly soil temperature profiles under the Control scenario (a), and the differences in soil temperature between the scenarios: Control-NoSurf (b) and Control-NoConv (c). Control, NoSurf and NoConv represent a full, partial and completely-absent consideration of convective heat transfer (CHT) in the SHAW model, respectively. NoSurf removes CHT due to infiltration and NoConv removes all CHT processes from the model.**

**RC2:** 10.What is the spring meltwater referred to in this article? If it is snow melt, how much snow is there on the Tibetan? How much melt water is there and how much of it is infiltrated into the soil, all these questions are not answered in the article.

**Authors:** The spring meltwater mentioned in this study included two parts, snowmelt water and ground ice melt water during spring. According to nearby observations (Xiao et al., 2013), the maximum snow depth of the surrounding of TGL site is about 22 cm among our study periods, and the days with snow depth below 5 cm accounts for 72% of all snow days. A short description to the site's snow condition was included in the revised manuscript:

*"According to continuous snow depth monitoring by an SR-50 ultrasonic snow depth sensor, the instantaneous maximum snow depth in the vicinity of the TGL site is about 22 cm, and the days with snow depth below 5 cm account for 72% of all snow days (Xiao, Zhao, and Li et al., 2013)"*

We have distinguished these two meltwater sources and have labeled the volume of the snow melt on the revised Figure 6b and 7b (marked by purple dots). We can see that the downward liquid flows at shallow depths were linked to snowmelt, indicating that snowmelt infiltration is one of the main components of downward flow. However,

infiltrations were not all from snowmelt. The simulated snowmelt in our study is about 190 mm per year, while the downward flow is about 220 mm at 0.05 m depth. The liquid water from the ground ice melting is the other important component that causes CHT events. When snow melt water infiltrated into the soil and mixed with soil's own moisture, it is hard to distinguish what fraction of the flux came from snowmelt and what fraction came from ground ice melt. So that we have to use the liquid flux instead of snowmelt volume to analyze the relationship between soil water migration and CHT.

Descriptions about snowmelt have been added at Section 3.3.1:

*"The downward flows are related to snowmelt events as shown in Figure 6b and Figure 7b, where only those simulated under Control are shown because the snowmelt events under the three scenarios are nearly identical. It indicates that infiltration of meltwater from snow is the major source of downward liquid flow during spring. Nevertheless, some of the liquid flux also came from ground ice melt, and it is difficult to distinguish what fraction of the flux came from snowmelt and what fraction came from ground ice melt. Thus, we used the total liquid flows instead of snowmelt volume to examine the relationship between soil water migration and CHT."*

The new Figure 6 and 7 also are presented here for your reference:

[Figure]

**Figure 6   Hourly soil temperature, water flux and UWC at 0.05 m depth, representative of shallow depths, simulated under NoSurf and Control during the 2008-2010 thaw periods. From top to bottom are: (a) the differences in soil temperature (T) between Control and NoSurf (Control-NoSurf), with positive values indicating heating effects and negative values indicating cooling effects; (b) snowmelt water simulated under Control and the water fluxes (WF) at 0.05 m simulated under NoSurf and Control, where positive values represent downward flows and negative for upward flows; (c) soil temperatures and (d) UWCs simulated under NoSurf and Control.**

[Figure]

**Figure 7** **Hourly soil temperature, water flux, and UWC at 0.05 m depth simulated under NoConv and Control during the 2008-2010 thaw periods. From top to bottom are: (a) the differences in soil temperature (T) between Control and NoConv (Control-NoConv), with positive values indicating heating effects and negative values indicating cooling effects; (b) snowmelt water simulated under Control and the water fluxes (WF) simulated under NoConv and Control, where positive values represent downward flows and negative for upward flows; (c) soil temperatures and (d) UWCs simulated under NoConv and Control.**

In order to help understand air temperature and precipitation/snowfall variations over the study period, we also provided new Figure 1 in the revised manuscript presenting daily air temperature and precipitation during 2008 to 2010 at the Tanggula site, and we state in the texts:

*"In the SHAW model, precipitation is assumed to be snowfall when air temperature is below 0 °C."*

[Figure]

**Figure 1** **Times series of daily air temperature at 2 m height and precipitation at the Tanggula (TGL) site during 2008-2010 aggregated from the hourly data that used in this study.**

**Reference:**

Xiao, Y., Zhao, L., Li, R., Jiao, K., Qiao, Y., Yao, J.: The Evaluation of Sr-50 for Snow Depth Measurements at Tanggula Area, Journal of Applied Meteorological Science, 24, 342-348, 2013.

---

## Author Response (AR2)

We would like to thank the reviewers and the Editor Ylva Sjöberg for their attention on our study and kind comments that help us improve the quality of our manuscript. Following the comments from the Referee, we have further revised the manuscript and we believe the quality has been further improved. Below please find the point-by-point responses to the comments.

**Texts in red** are the reviewer's comments; **those in black** are the authors' explanations to the reviewer's comments; and **those in blue** are the revised texts in the revised manuscript.

Comments from Referee #2:
I basically satisfied with your revisions on the manuscript. Please address following minor issues before publication.

(1) Some of its details are not adequately rigorous in the introduction, such as the results of Zhang et al. (2021) (Impact process and mechanism of summertime rainfall on thermal–moisture regime of active layer in permafrost regions of central Qinghai–Tibet Plateau), which also focus on the convective heat transfer. Authors are suggested to add a comprehensive summary of the latest researches and point out what the strengths of your findings are.

We are very grateful to the reviewer for updating us on relevant researches that we have not been aware of before. We have reviewed more recent works related to this topic. In this revision, we have added some representative researches published recently to the Introduction:
"As a result, warming of soil temperature by liquid CHT due to summertime rainfall increases the thaw depth of frozen ground (Douglas et al., 2020; Guan et al., 2010; Karjalainen et al., 2019) and promotes the greenhouse gas emissions (Neumann et al., 2019). However, an opposite view also exists that infiltration of precipitation has a cooling effect on the temperature of the active layer (Wen et al., 2014; Yang et al., 2018), indicating a complex mechanism of the CHT impacts on the soil thermal regime. "
and:
"However, relatively few studies have examined liquid CHT processes in permafrost context. Kurylyk et al. (2016) developed a three-dimensional coupled soil heat and water model to investigate the effects of runoff on soil temperature. Recently, Zhang, M. et al. (2021)quantified the energy flux of infiltrative CHT during a summertime rainfall event and reported that the thermal impacts of CHT were not pronounced compared to other energy transfer pathways. While their studies improve our understanding on the role of CHT in altering permafrost thermal dynamics, they focused on specific permafrost conditions or single events, and the established methods were difficult to transfer to other regions with conditions dissimilar to those in these study regions."
and to Discussion 4.1:

"The same weak significance of CHT associated with summer rainfall is also reported by Zhang, M. et al. (2021), where the positive energy flux of rainfall convection into soil layers was low, in contrast to other negative energy fluxes due to increased soil evaporation and latent heat, and decreased soil conductivity due to growing soil moisture."

In our study, A control experiment with three scenarios with full, partial and no consideration of CHT was implemented by the SHAW model. This design enables us to distinguish the CHT from different sources, i.e., precipitation/snowmelt and ground ice melt, and precisely quantify the impacts of these CHT processes, which has never been accomplished in previous studies that focused on reginal specific conditions or single summer rainfall events.

We have strengthened the description on the advantages in the last paragraph of Introduction:
"Therefore, this study uses the SHAW model to quantify the impacts of liquid CHT on the soil temperature and moisture in the active layer through numerical modelling at a typical permafrost site, i.e., the Tanggula (TGL) site on the QTP, China. The SHAW model was modified to exclude the CHT processes, and then control experiments were implemented to simulate comparative scenarios with or without CHT included in the model. This enables precise and separate quantification of the thermal impacts of liquid CHT from different sources such as precipitation infiltration, snow melt and ground ice melt, which has never been accomplished in previous studies. The specific objectives are: (1) to illustrate the characteristics of CHT events in time and depth; (2) to quantify the sole impacts of liquid CHT on the thermal regime of the active layer; (3) to elucidate the interplay of heat and soil moisture during the freezing-thawing process in the active layer. "

(2) It is recommended to add the simulation results of QT08 to the context for comparative analysis.
It's really a hard decision. Initially we don't want to include it in the manuscript because the data for QT08 simulation is not ideal and the results are not accurate enough to show the CHT effects. To accomplish the purpose of this study, so far only the TGL site on the QTP can provide sufficient supports of data and parameters. But as Reviewer said, a simulation at another permafrost site with success in reproducing the same CHT effects can help eliminate the worry that the findings from TGL are special for the TGL. Therefore, we decided to include the QT08 simulation.
In order to avoid messing up the structure of the manuscript, we chose to include the results of QT08 in Appendix A. We also discussed this comparative analysis in Discussion 4.1.

[revised manuscript text omitted]

(3) In table1: I noticed great difference between these soil texture data and the soil texture data for this site in other articles (especially the publication of co-author Dr. Lin Zhao's group), what is the reason for this?

The soil texture data at each soil depth applied in this study were based on the borehole information at TGL, which was published in a research article (in Chinese) in 2013 by Prof. Lin Zhao's group (Liu et al., 2013), as shown in Figure S1 below. After that, they subdivided the soil column and made more strata to fit a numerical simulation purpose (Hu et al. 2015). During this process, they made some adjustments, resulting in some differences between the original data and the modified data as shown in Figure S2. For example, the sand percentages at 0.1-0.2 m and around 2.45 m in the 2015's data are lower than in 2013's, and, oppositely, the percentages of slit become higher (Figure S1 and Figure S2). But basically, in both datasets the soil is mostly composed of sand (over 65% in both datasets) and the differences are not much.

We discussed this with Prof. Zhao and Dr. Hu, and we all agreed to use the original data from Liu et al. (2013) for this study. First, the 2015 data are not the original and have been purposely modified by some interpolation method. Second, we are modeling the period of 2008-2010, closer in time to Liu et al. (2013). In addition, we have applied an uncertainty

analysis for the key soil hydraulic parameters and then obtained the most optimal parameter values, which can also mitigate the influences of the given soil texture data.

表 2  活动层土壤各层土壤状况

Table 2  Soil information within the active layer at different depths

| Number 层次 | Depth 深度/m | Bulk density 土壤容重 /(g·cm⁻³) | Soil texture 土壤质地% | | |
|---|---|---|---|---|---|
| | | | 砂土 Sand | 黏土 Clay | 粉土 Silt |
| 1 | 0.0 | 1.176 | 93 | 6 | 1 |
| 2 | 0.02 | 1.176 | 93 | 6 | 1 |
| 3 | 0.05 | 1.176 | 93 | 6 | 1 |
| 4 | 0.1 | 1.176 | 93 | 6 | 1 |
| 5 | 0.2 | 1.331 | 87 | 10 | 3 |
| 6 | 0.5 | 1.103 | 89 | 9 | 2 |
| 7 | 0.7 | 1.405 | 87 | 10 | 3 |
| 8 | 0.9 | 1.405 | 84 | 13 | 3 |
| 9 | 1.05 | 1.235 | 75 | 18 | 7 |
| 10 | 1.4 | 1.281 | 75 | 18 | 7 |
| 11 | 1.75 | 1.253 | 71 | 21 | 8 |
| 12 | 2.1 | 1.460 | 71 | 21 | 8 |
| 13 | 2.45 | 1.332 | 71 | 21 | 8 |
| 14 | 2.8 | 1.109 | 71 | 21 | 8 |
| 15 | 3.0 | 1.832 | 71 | 21 | 8 |

**Figure S1 Soil texture at TGL site published in Liu et al., 2013**

Table 2  Soil texture parameters used as inputs for COUPMODEL

| Soil depth (cm) | Sand (%) | Silt (%) | Clay (%) | Soil depth (cm) | Sand (%) | Silt (%) | Clay (%) |
|---|---|---|---|---|---|---|---|
| 0–2 | 85 | 10 | 5 | 49–83 | 85 | 10 | 5 |
| 2–5 | 85 | 10 | 5 | 83–138 | 95 | 3 | 2 |
| 5–9 | 75 | 18 | 7 | 138–230 | 90 | 5 | 5 |
| 9–17 | 70 | 18 | 12 | 230–380 | 68 | 20 | 12 |
| 17–29 | 65 | 22 | 13 | 380–628 | 95 | 3 | 2 |
| 29–49 | 85 | 10 | 5 | – | – | – | – |

**Figure S2 Soil texture at TGL site published in Hu et al., 2015**

We stratified the soil layers into shallow, intermediate and deep depths, based on both the measurement depths in the active layer and the characteristics of the CHT impacts along the depth. On one hand, the soil column stratification scheme was designed according to the measurement depths of the active layer at TGL site. The near-surface soil was densely discretized with more depths (such as 0.00m, 0.02 m) to accommodate rapid hourly temperature and moisture dynamics near the ground surface. Because we don't have instruments installed at those depths, we used the first available depth (0.05m) to represent the shallow depths.

On the other hand, from Figure 4 in the revised manuscript and the supplementary Figure S3 shown below, we found that the CHT process have very non-uniform thermal effects between different soil depths, such as between 0.05 m, 1.05 m and 2.45 m. Therefore, it is necessary to explore how and why the CHT performs inconsistent at these different depths. We selected the 0.05 m depth, the closest depth to the ground surface and for which observations are available, for representing the shallow layers (above 0.2 m) which control by strong and rapid infiltrative CHT; and 1.05 m, which is at the middle depth of the entire active layer, for representing the intermediate layers (about 0.4 m to 1.3 m), where the infiltrative CHT has been weakened and CHT within the soil layers becomes more pronounced; and 2.45 m, adjacent to the bottom of the active layer, for representing the deep layers where no significant CHT process occurs.

We have added a new paragraph to justify the selection of those representative depths for our analyses at the beginning of Section 3.3:

"As shown in Figure 4a, CHT processes have very non-uniform thermal effects at different soil depths. Therefore, to illustrate the inconsistent impacts of CHT at depth and to identify the driving factors, three specific soil layers, i.e., the layer centered at 0.05 m depth closest to the ground surface and for which observations are available, the layer at 1.05 m depth, which is at the middle depth of the entire active layer, and the layer at 2.45 m adjacent to the bottom of the active layer, were selected to represent the shallow (0-0.2 m), intermediate (0.4-1.3 m) and deep depths (deeper than 1.75 m), respectively, where very different impacts of CHT were observed. "

[Figure]

**Figure S3 The differences in soil temperature between the scenarios: Control−NoSurf (right panels) and Control−NoConv (left panels) at 0.05 m, 0.1 m, 0.4 m, 1.05 m and 2.45 m. The temperature deviations present in this figure is the same as in Figure 4 b and c in the revised manuscript at the same depths.**

(5) What does it mean ground ice melt water during spring? Could the author depict the process of ground ice thawing when the active layer is completely frozen in spring?

We did not mean to express the process of ground ice thawing when the active layer is completely frozen in spring. Sorry for the misunderstanding caused here. Here, we want show is a simple top down thawing process during thaw periods.

It is shown in the Figure 6 and Figure 7 that in addition to the infiltration from ground surface, the melt water of ground ice in the active layer is also one contributor to the liquid migration fluxes among soils in thaw periods. Water flux from soil ice melt must happen after the upper layer thaws. We can see that the deeper soil layers always thawed later than the upper layers in Figure 4, and the CHT impacts of deeper layers also happened later than the upper layers, because the upper infiltrative water cannot percolate to the lower layers when the lower layers were still frozen. Thus, our simulation results do follow the principle that active layer thaws from top to bottom and the melt water are not able to transfer to the soil depth which is still frozen. To avoid any confusion here, we explained this top down ground ice thawing process in Section 3.3.1:

"Nevertheless, as the ambient temperature rises, the underlying frozen ground begins to thaw to depth (Figure 4b), and ground ice melt also partially contributes to the liquid flux. It is difficult to distinguish which fraction of the flux comes from snowmelt and which from ground ice melt."

(6) The simulation errors for soil temperature and moisture in the article are still large, especially below 0.4 m in Figure 2. So the possible reasons are suggested to discuss in the context.

Error sources in this simulation mainly include: (1) model structure uncertainty; (2) model parameter uncertainty, and (3) input data biases. We have performed a parameter uncertainty analysis for the SHAW simulation at TGL site and optimized the parametric values, so that the errors associated with parameter uncertainty can be much reduced. We also have discussed the flaws of model physical processes in simulating vapor flux direction and unfrozen water migration in freeze periods in Section 4.4.

According to previous studies, the results of SHAW are very sensitive to the lower boundary conditions. We found that the observed soil moisture data contain some abnormal zero values at and below 0.4 m depth. It is possible that the lower boundary conditions at 2.8 m that we applied to drive the model also lead to some uncertainties in the simulation results.

We provided an explanation related to the inaccuracy of lower boundary in Section 4.4:
"Therefore, in the TGL application, the inaccurate lower boundary conditions for the

SHAW model, particularly soil moisture, which is subject to appreciable measurable errors, also adversely affected the accuracy of the simulation at depth."

In addition to the changes listed above, we provided a new Figure 1b in the revised manuscript for presenting the locations of TGL and QT08 along with the permafrost distribution on Qinghai-Tibet Plateau. We have also improved the DPI of the images to 600 to make them clearer for reading. The inappropriate formatting in the reference list has also been fixed.

[Figure]

**Figure 1    Air temperature and precipitation (a) at the Tanggula (TGL) site during 2008-2010 and the map (b) showing the locations of permafrost sites considered in this study. Times series of daily air temperature at 2 m height and precipitation at TGL aggregated from the hourly measurements. The base map of permafrost distribution on the Qinghai-Tibet plateau (QTP) is from (Zou et al., 2017).**

---

## Author Response (AR3)

Dear Professor and Editor Ylva Sjöberg,

Thank you very much for your comments. In response to your comments, we have made further improvements to the manuscript. Below please find the point-by-point responses to your comments. Please refer to the change-tracked doc for all we have modified.

Thank you for submitting a revised version of your manuscript. I find your manuscript very close to publication in TC, but have a few minor and technical comments that I would like to see addressed before publication (line numbers refer to your tracked-changes ms version):

First of all, I'm missing a clear description of how the temperature of infiltrating water is determined, and in general how the top boundary is defined and potentially differ between the three models. (e.i. how is the temperature of snowmelt water and rain water determined?) This is also needed to understand the effects of the melt events in January 2010 in the three models. I'm asking for a brief description of these definitions.

As per your comment, we clarified the definitions of the temperature of water fluxes in Section 2.1:

"The SHAW model assumes that the migrating liquid and vapor water fluxes have the same temperature as the layers in which they are generated. Since the model does not provide a specific estimate of rain temperature and ignores the CHT processes within the canopy layer, rainwater entering the residue layer through the canopy is simply assumed to be at the same temperature as the residue layer when no snow cover is present on the surface. When snow is present, rainwater flowing through the canopy will participate in snow processes before it reaches the residue layer, and the temperature of snowmelt is assumed to be the same as the temperature of the snow layer at the time of melting."

**We also have highlighted the differences in the model configurations between the three scenarios in Section 2.2:**

"The resulting differences between the NoSurf/NoConv and Control simulations, each configured by the same model settings including the same meteorological forcing data, lower boundary conditions, and calibrated parametric values, thus reflect the effects of liquid CHT on the active layer dynamics."

The top boundaries of the simulations were defined using the same set of the meteorological forcing data. All model simulations were performed with the same model configuration. The only difference between the three scenarios are the models (one is the original SHAW model and the other two are modified SHAW models with partial of complete CHT processes removed).

L240 add space before parenthesis "m(Xiao, Zhao, and Dai et al., 2013)" L251 (Fig. 1) move parenthesis in "is from (Zou et al., 2017)" to "is from Zou et al. (2017)". L288 add space before parenthesis "TGL site(Liu et al., 2013)" Corrected.

Section 2.4: Either use sub-headings (e.g. "2.4.1 Driving data") or completely remove the words in bold (e.g. "Driving data") from the beginning of each paragraph in this section. We chose to remove these words.

L397-403: Clarify how CHT is responsible for warming in this case (January 2010). It reads like it is heat conduction from snowmelt water that lead to temperature increase - but was that process not included in all simulations? Did any of the cases not include snowmelt water at all?

It's a little bit complicated here. In our contrasting experiments, no water migration processes are changed, so the snowmelt flow in all scenarios is simulated the same. A same amount of liquid water will be generated and percolate through the snow layer to the topmost soil layer (the specific soil layer at 0.00 m). The difference lies in that in the Control scenario, sensible convective heat due to snowmelt percolation was considered, while in the other two scenarios, convective heat flux due to the snowmelt water movement was not considered because the CHT process at the ground surface was removed from the model. In January 2010, although the soil was frozen and impermeable, the abnormal snowmelt caused liquid water to flow to the 0 m soil layer. As a consequence, in the Control scenario, sensible convective heat flux due to snowmelt percolation altered the temperature of the 0 m soil layer and then the temperature gradient there. Although snowmelt water could not further penetrate the frozen soil layer, these temperature perturbations in the 0 m layer were then transmitted to the shallow soil layers by heat conduction. However, in the NoSurf and NoConv scenarios, despite the same amount of snowmelt water was accumulated in the 0 m layer like in the Control, no convective heat transfer occurred and no subsequent heat conduction would occur. Although heat conduction is involved here, it is a result of convective heat transfer due to snowmelt. We recognize the resulting differences in the shallow soil layers as the indirect effects of CHT on the ground surface. The resulting differences are specially distinct during January 2010. As this is an interesting finding, we provided detailed discussion on this in Section 4.2.

Accordingly, we have revised the relevant texts to clarity this indirect effect and the difference between the Control and the other two scenarios in January 2010 in Section 3.2. Hopefully, it is now clearer:

"The differences in soil temperature were noticeable even at shallow depths in January 2010 (Figure 4b and c), when soils at those depths were frozen and impermeable. This phenomenon coincided with the occurrence of extra snowmelt events during this period, as shown in Figure 3. Although snowmelt did not infiltrate into the underlying impermeable soil layers, it moved downward into the uppermost soil layer (0.00 m) during these periods. In the Control scenario, the sensible convective heat flux due to percolation of snowmelt water altered the temperature of the uppermost soil layer and consequently the temperature gradient there. These temperature perturbations were then transmitted to the near-surface

soil layers by conduction. In the other two scenarios, where the CHT process at the ground surface were excluded from the model, the same amount of snowmelt water was transported to the top soil layer, but no convective heat was transferred and thus no thermal disturbance occurred in the shallow soil layers as in the Control, as manifested in the temperature deviations in the shallow layers when contrasting the scenarios. It suggests that CHT could also have indirect thermal impacts during freeze periods, providing that snowmelt occurs during these periods in response to changes in air temperature."

L482: What do you mean by "pikes"? (peaks?) We replaced "pikes" with "peaks".

L491: This sentence needs some revision "Liquid water migration agrees the occurrence of CHT very well (Figure 6a and Figure 7a)". ("Liquid water migration correlates with the occurrence of CHT very well (Figure 6a and Figure 7a)"?) We used "correlates" instead in this revision.

L677: add space "Zweigel et al.(2021)have" Done

L757: Missing a word in this sentence? "During the spring thaw period in, the differences..."

We deleted the word "in".

I look forward to receiving a new version after these minor revisions.

Thank you very much!